# Diffusion-TTA: Test-time Adaptation of Discriminative Models via Generative Feedback

**Mihir Prabhudesai*  Tsung-Wei Ke*  Alexander C. Li  Deepak Pathak  Katerina Fragkiadaki**
{mprabhud,tsungwek,acl2,dpathak,katef}@cs.cmu.edu
Carnegie Mellon University

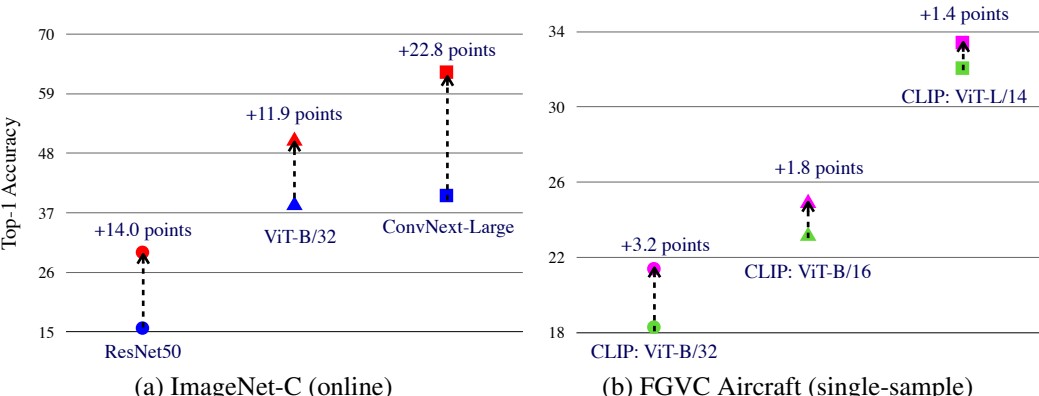

(a) ImageNet-C (online)  (b) FGVC Aircraft (single-sample)

Figure 1: **Diffusion-TTA improves state-of-the-art pre-trained image classifiers and CLIP models across various benchmarks.** Our model adapts pre-trained image discriminative models using feedback from pre-trained image generative diffusion models. *Left:* Image classification performance of pre-trained image classifiers improves after online adaptation. The image classifiers are pre-trained on ImageNet and adapted on ImageNet-C in an unsupervised manner using generative feedback. As can be seen, we get a significant boost across various model architectures. *Right:* Accuracy of open-vocabulary CLIP classifiers improves after single-sample adaptation, where we adapt to each unlabelled sample in the test set independently. CLIP is trained on millions of image-text pairs collected from the Internet [38], here we test it on the FGVC dataset [30].

## Abstract

The advancements in generative modeling, particularly the advent of diffusion models, have sparked a fundamental question: how can these models be effectively used for discriminative tasks? In this work, we find that generative models can be great test-time adapters for discriminative models. Our method, Diffusion-TTA, adapts pre-trained discriminative models such as image classifiers, segmenters and depth predictors, to each unlabelled example in the test set using generative feedback from a diffusion model. We achieve this by modulating the conditioning of the diffusion model using the output of the discriminative model. We then maximize the image likelihood objective by backpropagating the gradients to discriminative model's parameters. We show Diffusion-TTA significantly enhances the accuracy of various large-scale pre-trained discriminative models, such as, ImageNet classifiers, CLIP models, image pixel labellers and image depth predictors. Diffusion-TTA outperforms existing test-time adaptation methods, including TTT-MAE and TENT, and particularly shines in online adaptation setups, where the discriminative model is continually adapted to each example in the test set. We provide access to code, results, and visualizations on our website: diffusion-tta.github.io.

---

*Equal Technical Contribution

37th Conference on Neural Information Processing Systems (NeurIPS 2023).

# 1 Introduction

Currently, all state-of-the-art predictive models in machine learning, that care to predict an output $y$ from an input $x$, are discriminative in nature, that means, they are functions trained to map directly $x$ to (a distribution of) $y$. For example, existing successful image understanding models, whether classifiers, object detectors, segmentors, or image captioners, are trained to encode an image into features relevant to the downstream task through end-to-end optimization of the end objective, and thus ignore irrelevant details. Though they shine within the training distribution, they often dramatically fail outside of it [14, 25], mainly because they learn shortcut pathways to better fit $p(y|x)$. Generative models on the other hand are trained for a much harder task of modeling $p(x|y)$. At test-time, these models can be inverted using the Bayes rule to infer the hypothesis $\hat{y}$ that maximizes the data likelihood of the example at hand, $p(x|\hat{y})$. Unlike discriminative models that take an input and predict an output in a feed-forward manner, generative models work backwards, by searching over hypotheses $y$ that fit the data $x$. This iterative process, also known as analysis-by-synthesis [52], is slower than feed-forward inference. However, the ability to generate leads to a richer and a more nuanced understanding of the data, and hence enhances its discriminative potential [22]. Recent prior works, such as Diffusion Classifier [27], indeed find that inverting generative models generalizes better at classifying out-of-distribution images than popular discriminative models such as ResNet [17] and ViT [11]. However, it is also found that pure discriminative methods still outperform inversion of generative methods across almost all popular benchmarks, where test sets are within the training distribution. This is not too surprising since generative models aim to solve a much more difficult problem, and they were never directly trained using the end task objective.

In this paper, instead of considering generative and discriminative models as competitive alternatives, we argue that they should be coupled during inference in a way that leverages the benefits of both, namely, the iterative reasoning of generative inversion and the better fitting ability of discriminative models. A naive way of doing so would be to ensemble them, that is $\frac{p_\theta(y|x)+p_\phi(y|x)}{2}$, where $\theta$ and $\phi$ stand for discriminative and generative model parameters, respectively. However, empirically we find that this does not result in much performance boost, as the fusion between the two models happens late, at the very end of their individual predictions. Instead, we propose to leverage generative models to adapt discriminative models at test time through iterative optimization, where both models are optimized for maximizing a sample's likelihood.

We present Diffusion-based Test Time Adaptation (TTA) (Diffusion-TTA), a method that adapts discriminative models, such as image classifiers, segmenters and depth predictors, to individual unlabelled images by using their outputs to modulate the conditioning of an image diffusion model and maximize the image diffusion likelihood. This operates as an inversion of the generative model, to infer the discriminative weights that result in the discriminative hypothesis with the highest conditional image likelihood. Our model is reminiscent of an encoder-decoder architecture, where a pre-trained discriminative model encodes the image into a hypothesis, such as an object category label, segmentation map, or depth map, which is used as conditioning to a pre-trained generative model to generate back the image. We show that Diffusion-TTA effectively adapts image classifiers for both in- and out-of-distribution examples across established benchmarks, including ImageNet and its variants, as well as image segmenters and depth predictors in ADE20K and NYU Depth dataset.

Generative models have previously been used for test time adaptation of image classifiers and segmentors, by co-training the model under a joint discriminative task loss and a self-supervised image reconstruction loss, e.g., TTT-MAE [12] or Slot-TTA [37]. At test time, the discriminative model is finetuned using the image reconstruction loss. While these models show that adaptation boosts performance, this boost often stems from the subpar performance of their initial feed-forward discriminative model. This can be clearly seen in our TTT-MAE baseline [12], where the before-adaptation results are significantly lower than a pre-trained feed-forward classifier. In contrast, our approach refrains from any joint training, and instead directly adapts *pre-trained* discriminative models at test time using pre-trained generative diffusion models.

We test our approach on multiple tasks, datasets and model architectures. For classification, we test pre-trained ImageNet clasisifers on ImageNet [9], and its out-of-distribution variants (C, R, A, v2, S). Further we test, large-scale open-vocabulary CLIP-based classifiers on CIFAR100, Food101, FGVC, Oxford Pets, and Flowers102 datasets. For adapting ImageNet classifiers we use DiT [36] as our generative model, which is a diffusion model trained on ImageNet from scratch. For adapting open-vocabulary CLIP-based classifiers, we use Stable Diffusion [40] as our generative model. We

show consistent improvements over the initially employed classifier as shown in Figure 1. We also test on the adaptation of semantic segmentation and depth estimation tasks, where segmenters of SegFormer [49] and depth predictors of DenseDepth [1] performance are greatly improved on ADE20K and NYU Depth v2 dataset. For segmentation and depth prediction, we use conditional latent diffusion models [40] that are trained from scratch on their respective datasets. We show extensive ablations of different components of our Diffusion-TTA method, and present analyses on how diffusion generative feedback enhances discriminative models. We hope our work to stimulate research on combining discriminative encoders and generative decoder models, to better handle images outside of the training distribution. Our code and trained models are publicly available in our project's website: `diffusion-tta.github.io/`.

## 2 Related Work

**Generative models for discriminative tasks**   Recently there have been many works exploring the application of generative models for discriminative tasks. We group them into the following three categories: **(i) Inversion-Based Methods**: Given a test input $x$ and a conditional generative model $p_\phi(x|\mathbf{c})$, these methods make a prediction by finding the conditional representation $\mathbf{c}$ that maximizes the estimated likelihood $p_\phi(x|\mathbf{c})$ of the test input. Depending on the architecture of the conditional generative model, this approach can infer the text caption [8, 27], class label [27], or semantic map [7] of the input image. For instance, Li et al. [27] showed that finding the text prompt that maximizes the likelihood of an image under a text-to-image model like Stable Diffusion [40] is a strong zero-shot classifier, and finding the class index that maximizes the likelihood of an image under a class-conditional diffusion model is a strong standard classifier. **(ii) Generative Models for Data Augmentation:** This category of methods involves using a generative model to enhance the training data distribution [2, 45, 51]. These methods create synthetic data using the generative model and train a discriminative classifier on the augmented training dataset. For instance, Azizi et al. [2] showed that augmenting the real training data from ImageNet with synthetic data from Imagen [41], their large-scale diffusion model, improves classification accuracy. **(iii) Generative Models as feature extractors**: These techniques learn features with a generative model during pretraining and then fine-tune them on downstream discriminative tasks [3, 10, 18, 35, 42, 50]. Generative pre-training is typically a strong initialization for downstream tasks. However, these methods require supervised data for model fine-tuning and cannot be directly applied in a zero-shot manner.

Our work builds upon the concept of Inversion-Based Methods (i). We propose that instead of inverting the conditioning representation, one should directly adapt a pre-trained discriminative model using the likelihood loss. This strategy enables us to effectively combine the best aspects of both generative and discriminative models. Our method is complementary to other methods of harnessing generative models, such as generative pre-training or generating additional synthetic labeled images

**Test-time adaptation**   Test-time adaptation (TTA) also referred as unsupervised domain adaptation (UDA) in this context, is a technique that improves the accuracy of a model on the target domain by updating its parameters without using any labelled examples. Methods such as pseudo labeling and entropy minimization [46] demonstrate that model accuracy can be enhanced by using the model's own confident predictions as a form of supervision. These methods require batches of examples from the test distribution for adaptation. In contrast, TTA approaches based on self-supervised learning (SSL) have demonstrated empirical data efficiency. SSL methods jointly train using both the task and SSL loss, and solely use the SSL loss during test-time adaptation [5, 12, 16, 31, 37, 44]. These methods do not require batch of examples and can adapt to each example in the test set independently. Our contribution falls within the SSL TTA framework, where we use diffusion generative loss for adaptation. We refrain from any form of joint training and directly employ a pre-trained conditional diffusion model to perform test-time adaptation of a pre-trained discriminative model.

## 3 Test-Time Adaptation with Diffusion Models

The architecture of Diffusion-TTA method is shown in Figure 2 and its pseudocode in shown in Algorithm 1. We discuss relevant diffusion model preliminaries in Section 3.1 and describe Diffusion-TTA in detail in Section 3.2.

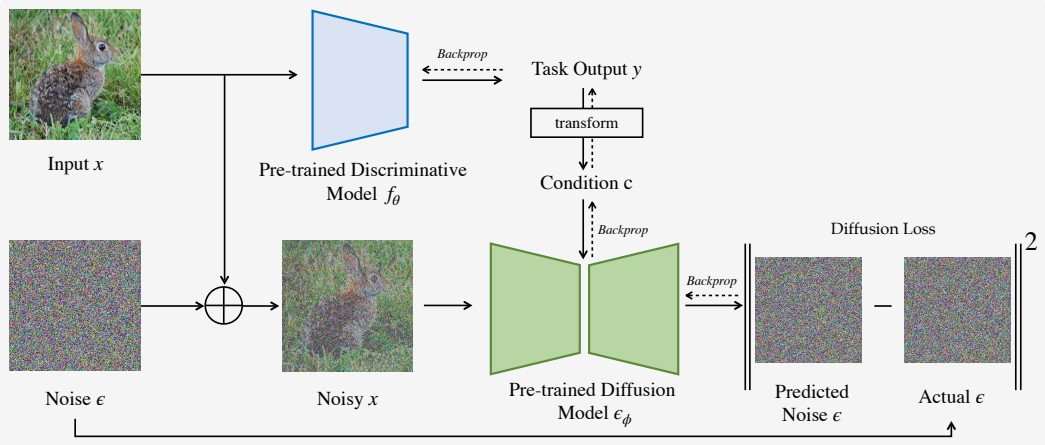

Figure 2: **Architecture of Diffusion-TTA.** Our method consists of discriminative and generative modules. Given an image $x$, the discriminative model $f_\theta$ predicts task output $y$. The task output $y$ is transformed into condition $\mathbf{c}$. Finally, we use the generative diffusion model $\epsilon_\phi$ to measure the likelihood of the input image, conditioned on $\mathbf{c}$. This consists of using the diffusion model $\epsilon_\phi$ to predict the added noise $\epsilon$ from the noisy image $x_t$ and condition $\mathbf{c}$. We maximize the image likelihood using the diffusion loss by updating the discriminative and generative model weights via backpropagation. For a task-specific version of this figure, please refer to Figure 7 in the Appendix.

## 3.1 Diffusion Models

A diffusion model learns to model a probability distribution $p(x)$ by inverting a process that gradually adds noise to the image $x$. The diffusion process is associated with a variance schedule $\{\beta_t \in (0,1)\}_{t=1}^T$, which defines how much noise is added at each time step. The noisy version of sample $x$ at time $t$ can then be written $x_t = \sqrt{\bar{\alpha}_t}x + \sqrt{1-\bar{\alpha}_t}\epsilon$ where $\epsilon \sim \mathcal{N}(\mathbf{0},\mathbf{1})$, is a sample from a Gaussian distribution (with the same dimensionality as $x$), $\alpha_t = 1 - \beta_t$, and $\bar{\alpha}_t = \prod_{i=1}^t \alpha_i$. One then learns a denoising neural network $\epsilon_\phi(x_t;t)$ that takes as input the noisy image $x_t$ and the noise level $t$ and tries to predict the noise component $\epsilon$. Diffusion models can be easily extended to draw samples from a distribution $p(x|\mathbf{c})$ conditioned on a condition $\mathbf{c}$, where $\mathbf{c}$ can be an image category, image caption, semantic map, a depth map or other information [28, 40, 54]. Conditioning on $\mathbf{c}$ can be done by adding $\mathbf{c}$ as an additional input of the network $\epsilon_\phi$. Modern conditional image diffusion models are trained on large collections $\mathcal{D} = \{(x^i, \mathbf{c}^i)\}_{i=1}^N$ of $N$ images paired with conditionings by minimizing the loss:

$$\mathcal{L}_{\text{diff}}(\phi; \mathcal{D}) = \frac{1}{|\mathcal{D}|} \sum_{x^i, \mathbf{c}^i \in \mathcal{D}} ||\epsilon_\phi(\sqrt{\bar{\alpha}_t}x^i + \sqrt{1-\bar{\alpha}_t}\epsilon, \mathbf{c}^i, t) - \epsilon||^2. \qquad (1)$$

## 3.2 Test-time Adaptation with Diffusion Models

In Diffusion-TTA, the condition $\mathbf{c}$ of the diffusion model depends on the output of the discriminative model. Let $f_\theta$ denote the discriminative model parameterized by parameters $\theta$, that takes as input an image $x$ and outputs $y = f_\theta(x)$. We consider image classifiers, image pixel labellers or image depth predictors as candidates for discriminative models to adapt.

For image classifiers, $y$ represents a probability distribution over $L$ categories, $y \in [0,1]^L, y^\top \mathbf{1}_L = 1$. Given the learnt text embeddings of a text-conditional diffusion model for the $L$ categories $\ell_j \in \mathbb{R}^d, j \in \{1..L\}$, we write the diffusion condition as $\mathbf{c} = \sum_{j=1}^L y_j \cdot \ell_j$.

For pixel labellers, $y$ represents the set of per-pixel probability distributions over $L$ categories, $y = \{y^u \in [0,1]^L, y^{u\top}\mathbf{1}_L = 1, u \in x\}$. The diffusion condition then is $\mathbf{c} = \{\sum_{j=1}^L y_j^u \cdot \ell_j, u \in x\}$, where $\ell_j$ is the embedding of category $j$.

For depth predictors $y$ represents a depth map $y \in \mathbb{R}^{+w \times h}$, where $w, h$ the width and height of the image, and $\mathbf{c} = y$.

As $\mathbf{c}$ is differentiable with respect to the discriminative model's weights $\theta$, we can now update them via gradient descent by minimizing the following diffusion loss:

$$L(\theta, \phi) = \mathbb{E}_{t,\epsilon} \| \epsilon_\phi(\sqrt{\bar{\alpha}_t}x + \sqrt{1 - \bar{\alpha}_t}\epsilon, \mathbf{c}, t) - \epsilon \|^2. \tag{2}$$

We minimize this loss by sampling random timesteps $t$ coupled with random noise variables $\epsilon$. For online adaptation, we continually update the model weights across test images that the model encounters in a streaming manner. For single-sample adaptation, the model parameters are updated to each sample in the test set independently.

---

**Algorithm 1** `Diffusion-TTA`

---

1: **Input:** Test image $x$, discriminative model weights $\theta$, diffusion model weights $\phi$, adaptation steps $N$, batch size $B$, learning rate $\eta$.
2: **for** adaptation step $s \in (1, \ldots, N)$ **do**
3:     Compute current discriminative output $y = f_\theta(x)$
4:     Compute condition $\mathbf{c}$ based on output $y$
5:     Sample timesteps $\{t_i\}_{i=1}^B$ and noises $\{\epsilon_i\}_{i=1}^B$
6:     Loss $L(\theta, \phi) = \frac{1}{N} \sum_{i=1}^B \| \epsilon_\phi(\sqrt{\bar{\alpha}_{t_i}}x + \sqrt{1 - \bar{\alpha}_{t_i}}\epsilon_i, \mathbf{c}, t_i) - \epsilon_i \|^2$
7:     Update classifier weights $\theta \leftarrow \theta - \eta \nabla_\theta L(\theta, \phi)$
8:     Optional: update diffusion weights $\phi \leftarrow \phi - \eta \nabla_\phi L(\theta, \phi)$
9: **end for**
10: **return** updated output $y = f_\theta(x)$

---

**Implementation Details.** Our experiments reveal that minimizing the variance in the diffusion loss significantly enhances our test-time adaptation performance. We achieve this by crafting a larger batch size through randomly sampling timesteps from a uniform distribution, coupled with randomly sampling noise vectors from a standard Gaussian distribution.

We conduct our experiments on a single NVIDIA-A100 40GB VRAM GPU, with a batch size of approximately 180. Since our GPU fits only a batch size of 20, we utilize a gradient accumulation of nine steps to expand our batch size to 180. Consequently, this setup results in an adaptation time of roughly 55 seconds per example. We use Stochastic Gradient Descent (SGD) with a learning rate of 0.005 or Adam optimizer [24] with a learning rate of 0.00001. We set momentum to 0.9.

For all experiments, we adjust all the parameters of the discriminative model. We freeze the diffusion model parameters for the open-vocabulary experiments in Section 4.2 with CLIP and Stable Diffusion 2.0, and otherwise, update all diffusion model parameters. For CLIP, instead of adapting the whole network we only adapt the LoRA adapter weights [23] added to the query and value projection layers of the CLIP model. For our experiments on adapting to ImageNet distribution shift in Section 4.1, we observe a significant performance boost when adapting both the discriminative and generative model. This finding suggests that our method effectively combines the distinct knowledge encoded by these two models. We present a more detailed analyses in Section 4.4.

## 4 Experiments

We test Diffusion-TTA in adapting ImageNet classifiers [11, 17, 29], CLIP models [38], image pixel labellers [49], and depth predictors [1] across multiple image classification, semantic segmentation, and depth estimation datasets, for both in-distribution and out-of-distribution test images. Our experiments aim to answer the following questions:

1. How well does Diffusion-TTA test-time adapt ImageNet and CLIP classifiers in comparison to other TTA methods under online and single-sample adaptation settings?

2. How well does Diffusion-TTA test-time adapt semantic segmentation and depth estimation models?

3. How does performance of our model vary for in- and out-of-distribution images?

4. How does performance of our model vary with varying hyperparameters, such as layers to adapt, batchsize to use, or diffusion timesteps to consider?

**Evaluation metrics**. We report the top-1 accuracy, mean pixel accuracy, and Structure Similarity index [48] on classification, semantic segmentation, and depth estimation benchmark datasets respectively.

## 4.1 Test-time Adaptation of Pre-trained ImageNet Classifiers

We use Diffusion-TTA to adapt multiple ImageNet classifiers with varying backbone architectures and sizes: ResNet-18 [17], ResNet-50, ViT-B/32 [11], and ConvNext-Tiny/Large [29]. For fair comparisions, we use Diffusion Transformer (DiT-XL/2) [36] as our class-conditional generative model, which is trained on ImageNet from scratch.

**Datasets** We consider the following datasets for TTA of ImageNet classifiers: ImageNet [9] and its out-of-distribution counterparts: ImageNet-C [19] (level-5 gaussian noise), ImageNet-A [21], ImageNet-R [20], ImageNetV2 [39], and Stylized ImageNet [13].

**Baselines** We compare our model against the following state-of-the-art TTA approaches. We use their official codebases for comparision:

- TTT-MAE [12] is a per-image test-time adaptation model trained under a joint classification and masked autoencoding loss, and test-time adapted using only the (self-supervised) masked autoencoding loss. For comparison, we use the numbers reported in the paper when possible, else we test the publicly available model.
- TENT [46] is a TTA method that adapts the batchnorm normalization parameters by minimizing the entropy (maximizing the confidence) of the classifier at test time per example.
- COTTA [47] test-time adapts a (student) classifier using *pseudo labels*, that are the weighted-averaged classifications across multiple image augmentations predicted by a teacher model.

| | ImageNet | ImageNet-A | ImageNet-R | ImageNet-C | ImageNet-V2 | ImageNet-S |
|---|---|---|---|---|---|---|
| Customized ViT-L/16 classifier [12] | 82.1 | 14.4 | 33.0 | 17.5 | **72.5** | 11.9 |
| + TTT-MAE (single-sample) | 82.0 (-0.1) | 21.3 (+6.9) | 39.2 (+6.2) | 27.5 (+10.0) | 72.3 (-0.2) | 20.2 (+0.3) |
| ResNet18 | 69.5 | 1.4 | 34.6 | 2.6 | 57.1 | 7.7 |
| + TENT (online) | 63.0 (-6.5) | 0.6 (-0.8) | 34.7 (+0.1) | **12.1 (+9.5)** | 52.0 (-5.1) | 9.8 (+2.1) |
| + CoTTA (online) | 63.0 (-6.5) | 0.7 (-0.7) | 34.7 (+0.1) | 11.7 (+9.1) | 52.1 (-5.0) | 9.7 (+2) |
| + Diffusion-TTA (single-sample) | **77.2 (+7.7)** | **6.1 (+4.7)** | **39.7 (+5.1)** | 4.5 (+1.9) | **63.8 (+6.7)** | **12.3 (+4.6)** |
| ViT-B/32 | 75.7 | 9.0 | 45.2 | 39.5 | 61.0 | 15.8 |
| + TENT (online) | 75.7 (0.0) | 9.0 (0.0) | 45.3 (+0.1) | 38.9 (-0.6) | 61.1 (+0.1) | 10.4 (-5.4) |
| + CoTTA (online) | 75.8 (+0.1) | 8.6 (-0.4) | 45.0 (-0.2) | 40.0 (+0.5) | 60.9 (-0.1) | 1.1 (-14.7) |
| + Diffusion-TTA (single-sample) | **77.6 (+1.9)** | **11.2 (+2.2)** | **46.5 (+1.3)** | **41.4 (+1.9)** | **64.4 (+3.4)** | **21.3 (+5.5)** |
| ConvNext-Tiny | 81.9 | 22.7 | 47.8 | 16.4 | 70.9 | 20.2 |
| + TENT (online) | 79.3 (-2.6) | 10.6 (-12.1) | 42.7 (-5.1) | 2.7 (-13.7) | 69.0 (-1.9) | 19.9 (-0.3) |
| + CoTTA (online) | 80.5 (-1.4) | 13.2 (-9.5) | 47.2 (-0.6) | 13.7 (-2.7) | 68.9 (-2.0) | 19.3 (-0.9) |
| + Diffusion-TTA (single-sample) | **83.1 (+1.2)** | **25.8 (+3.1)** | **49.7 (+1.9)** | **21.0 (+4.6)** | **71.5 (+0.6)** | **22.6 (+2.4)** |

Table 1: **Singe-sample test-time adaptation of ImageNet-trained classifiers.** We observe consistent and significant performance gain across all types of classifiers and distribution drifts.

We present classification results on in-distribution (ImageNet) and out-of-distribution (ImageNet-C, R, A, V2, and S) for single-sample adaptation in Table 1. Further we present online adaptation results in Table 2. We evaluate on ImageNet-C, which imposes different types of corruption on ImageNet. For both COTTA and TENT we always report online-adaptation results as they are not applicable in the single-sample setting.

| ImageNet Corruption: | Gaussian Noise | Fog | Pixelate | Snow | Contrast |
|---|---|---|---|---|---|
| Customized ViT-L/16 classifier [12] | 17.1 | 38.7 | 47.1 | 35.6 | 6.9 |
| + TTT-MAE (online) | 37.9 (+20.8) | 51.1 (+12.4) | 65.7 (+18.6) | 56.5 (+20.9) | 10.0 (+3.1) |
| ResNet50 | 6.3 | 25.2 | 26.5 | 16.7 | 3.6 |
| + TENT (online) | 12.3 (+6.0) | **43.2 (+18.0)** | 41.8 (+15.3) | 28.4 (+11.7) | **12 (+8.4)** |
| + CoTTA (online) | 12.2 (+5.9) | 42.4 (+17.2) | 41.7 (+15.2) | 28.6 (+11.9) | 11.9 (+8.3) |
| + Diffusion-TTA (online) | **19.0 (+12.7)** | **43.2 (+18.0)** | **50.2 (+23.7)** | **33.6 (+16.9)** | 2.7 (-0.9) |
| ViT-B/32 | 39.5 | 35.9 | 55.0 | 30.0 | 31.5 |
| + TENT (online) | 38.9 (-0.6) | 35.8 (-0.1) | 55.5 (+0.5) | 30.7 (+0.7) | 32.1 (+0.6) |
| + CoTTA (online) | 40.0 (+0.5) | 34.6 (-1.3) | 54.5 (-0.5) | 29.7 (-0.3) | 32.0 (+0.5) |
| + Diffusion-TTA (online) | **46.5 (+7.0)** | **56.2 (+20.3)** | **64.7 (+9.7)** | **50.4 (+20.4)** | **33.6 (+2.1)** |
| ConvNext-Tiny | 16.4 | 32.3 | 37.2 | 38.3 | 32.0 |
| + TENT (online) | 2.7 (-13.7) | 5.0 (-27.3) | 43.9 (+6.7) | 15.2 (-23.1) | 40.7 (-18.7) |
| + CoTTA (online) | 13.7 (-2.7) | 29.8 (-2.5) | 37.3 (+0.1) | 26.6 (-11.7) | 32.6 (+0.6) |
| + Diffusion-TTA (online) | **47.4 (+31.0)** | **65.9 (+33.6)** | **69.0 (+31.8)** | **62.6 (+24.3)** | **46.2 (+14.2)** |
| ConvNext-Large | 33.0 | 34.4 | 49.3 | 44.5 | 39.8 |
| + TENT (online) | 30.8 (-2.2) | 53.5 (+19.1) | 51.1 (+1.8) | 44.6 (+0.1) | 52.4 (+12.6) |
| + CoTTA (online) | 33.3 (+0.3) | 15.1 (-18.7) | 34.6 (-15.3) | 7.7 (-36.8) | 10.7 (-29.1) |
| + Diffusion-TTA (online) | **54.9 (+21.9)** | **67.7 (+33.3)** | **71.7 (+22.4)** | **64.8 (+20.3)** | **55.7 (+15.9)** |

Table 2: **Online test-time adaptation of ImageNet-trained classifiers on ImageNet-C.** Our model achieves significant performance gain using online adaptation across all types of classifiers and distribution drifts.

Our conclusions are as follows:

(i) **Our method consistently improves classifiers** on in-distribution (ImageNet) and OOD test images across different image classifier architectures. For all ResNet, ViT, and ConvNext-Tiny, we observe significant performance gains.

(ii) **Diffusion-TTA outperforms TENT and COTTA across various classifier architectures.** Our results are consistent with the analysis in [55]: methods that primarily work under online settings (TENT or CoTTA) are not robust to different types of architectures and distribution shifts (see Table 4 in [55]).

(iii) **Diffusion-TTA outperforms TTT-MAE** even with a smaller-size classifier. ConvNext-tiny (28.5M params) optimized by Diffusion-TTA (online) achieves better performance than a much bigger custom backbone of ViT-L + ViT-B (392M params) optimized by TTT-MAE (online).

(iv) **Diffusion-TTA significantly improves ConvNext-Large**, in online setting. ConvNext-Large is the largest available ConvNext model with 197M parameters and achieves state-of-the-art performance on ImageNet.

### 4.2 Test-time Adaptation of Open-Vocabulary CLIP Classifiers

CLIP models are trained across millions of image-caption pairs using contrastive matching of the language and image feature embeddings. We test Diffusion-TTA for adapting three different CLIP models with different backbone sizes: ViT-B/32, ViT-B/16, and ViT-L/14. Experiments are conducted under single-sample adaptation setup.

We convert CLIP into a classifier following [38], where we compute the latent embedding for each text category label in the dataset using the CLIP text encoder, and then compute its similarity with the image latent to predict the classification logits across all class labels. Note that the CLIP models have not seen test images from these datasets during training.

**Datasets** We consider the following datasets for TTA of CLIP classifiers: ImageNet [9], CIFAR-100 [26], Food101 [6], Flowers102 [32], FGVC Aircraft [30], and Oxford-IIIT Pets [34] annotated with 1000,100,101, 102, 100, and 37 classes, respectively.

|  | Food101 | CIFAR-100 | Aircraft | Oxford Pets | Flowers102 | ImageNet |
|---|---|---|---|---|---|---|
| CLIP-ViT-B/32 | 82.6 | 61.2 | 17.8 | 82.2 | 66.7 | 57.5 |
| + Diffusion-TTA (Ours) | **86.2 (+3.6)** | **62.8 (+1.6)** | **21.0 (+3.2)** | **84.9 (+2.7)** | **67.7 (+1.0)** | **60.8 (+3.3)** |
| CLIP-ViT-B/16 | 88.1 | **69.4** | 22.8 | 85.5 | 69.3 | 62.3 |
| + Diffusion-TTA (Ours) | **88.8 (+0.7)** | 69.0 (-0.4) | **24.6 (+1.8)** | **86.1 (+0.6)** | **71.5 (+2.2)** | **63.8 (+1.5)** |
| CLIP-ViT-L/14 | **93.1** | 79.6 | 32.0 | 91.9 | 78.8 | 70.0 |
| + Diffusion-TTA (Ours) | **93.1** (0.0) | **80.6 (+1.0)** | **33.4 (+1.4)** | **92.3 (+0.4)** | **79.2 (+0.4)** | **71.2 (+1.2)** |

Table 3: **Single-sample Test-time adaptation of CLIP classifiers**. Our evaluation is performed across multiple model sizes and a variety of test datasets.

We show test-time adaptation results in Table 3. **Our method improves CLIP classifiers of different sizes consistently over all datasets**, including small-scale (CIFAR-100), large-scale (ImageNet), and fine-grained (Food101, Aircraft, Pets, and Flowers102) datasets.

| Corruption: | Clean | Gaussian-Noise | Fog | Frost | Snow | Contrast | Shot |
|---|---|---|---|---|---|---|---|
| Segmentation: SegFormer | 66.1 | 65.3 | 63.0 | 58.0 | 55.2 | 65.3 | 63.3 |
| + Diffusion-TTA | 66.1 (0.0) | **66.4 (+1.1)** | **65.1 (+2.1)** | **58.9 (+0.9)** | **56.6 (+1.4)** | **66.4 (+1.1)** | **63.7 (+0.4)** |
| Depth: DenseDepth | 92.4 | 79.1 | 81.6 | 72.2 | 72.7 | 77.4 | 81.3 |
| + Diffusion-TTA | **92.6 (+0.3)** | **82.1 (+3.0)** | **84.4 (+2.8)** | **73.0 (+0.8)** | **74.1 (+1.4)** | 77.4 | **82.1 (+0.8)** |

Table 4: **Single-sample test-time adaptation for semantic segmentation on ADE20K and depth estimation on NYU Depth v2.** We observe consistent and significant performance gain across all types of distribution drifts.

| Image | ground-truth | Before-TTA | After-TTA |
|---|---|---|---|

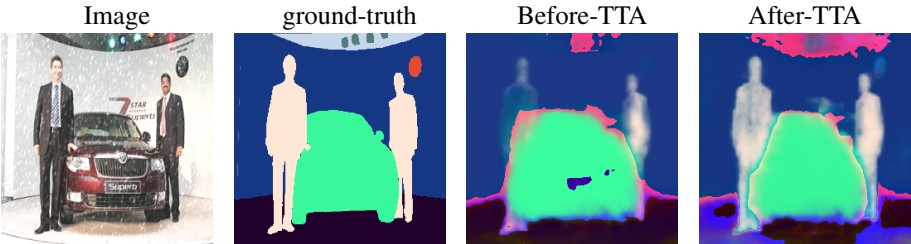

Figure 3: **Semantic segmentation before and after TTA.** We colorize ground-truth / predicted segmentation based on discrete class labels. Our method improves segmentation after adaptation.

### 4.3 Test-time Adaptation on Semantic Segmentation and Depth Estimation

We use Diffusion-TTA to adapt semantic segmentors of SegFormer [49] and depth predictors of DenseDepth [1]. We use a latent diffusion model [40] which is pre-trained on ADE20K and NYU Depth v2 dataset for the respective task. The diffusion model concatenates the segmentation/depth map with the noisy image during conditioning.

**Datasets** We consider ADE20K and NYU Depth v2 test sets for evaluating semantic segmentation and depth estimation. Note that both discriminative and generative diffusion models are trained on the same dataset. We evaluate with different types of image corruption under single-sample settings.

We show quantitative semantic segmentation and depth estimation results in Table 4. Further we show some qualitative semantic segmentation results before and after TTA in Figure 3. Our Diffusion-TTA improves both tasks consistently across all types of distribution drifts.

## 4.4 Ablations

We present ablative analysis of various design choices. We study how Diffusion-TTA varies with hyperparameters such as diffusion timesteps, number of samples per timestep and batchsize. We also study the effect of adapting different model parameters. Additionally we visualize before and after TTA adaptation results in Figure 5.

**Ablation of hyperparameters**  In Table 5, we ablate hyperparameters of our model on ImageNet-A dataset with ConvNext as our pre-trained classifier and DiT as our diffusion model. If we perform test-time adaptation using a single randomly sampled timestep and noise latent (*+diffusion TTA*), we find a significant reduction in the classification accuracy ($-2.9\%$). Increasing the batch size from 1 to 180 by sampling random timesteps from an uniform distribution (*+ timestep aug BS=180*) gives a significant boost in accuracy ($+4.8\%$). As shown in Figure 4, correct class labels do not always have lower loss across all timesteps, and thus using only single diffusion step is a bad approximation of the likelihood. Further sampling random noise latents per timestep (*+ noise aug*) gives an added boost of ($+0.2\%$). Finally, adapting the diffusion weights of DiT (*+ adapting diffusion weights*) in Section 4.1, gives further ($+1.1\%$) boost.

**Ablation of parameters/layers to adapt**  In Table 6, we ablate different paramters to adapt. We conduct our ablations on on ImageNet and ImageNet-R while using ResNet18 and DiT as our discriminative and generative models. We compare against following baselines: **1.** we randomly initialize the classifier instead of using the pre-trained weights **2.** we do not use any pre-trained classifier, instead we initialize the logits

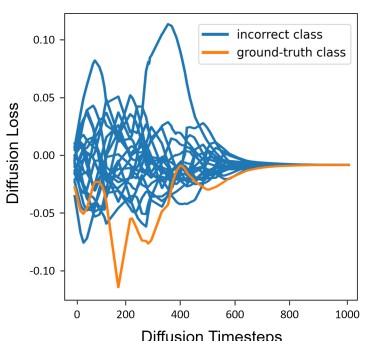

Figure 4: **Diffusion loss of class labels at each diffusion timestep**. The ground-truth class does not **always** have lower loss across all timesteps, compared to incorrect classes. Thus using only single diffusion timestep is a bad approximation of the likelihood.

using zeros and optimize them per example using diffusion objective (*adapt logits*), **3.** we average the probabilities of the *adapt logits* baseline with the probabilities predicted by ResNet18 (*adapt logits + ensemble*), **4.** we adapt only the batch normalization layers in ResNet18 (*adapt BN*), **5.** we adapt only the last fully connected layer in ResNet18 (*adapt last FC layer*), **6.** we adapt the whole ResNet18 (*adapt classifier*), and **7.** we adapt the whole ResNet18 and DiT, which refers to our method Diffusion-TTA. Adapting the whole classifier and generative model achieves the best performance. We randomly sample one image per category for the experiment.

|  | ImageNet-A |
|---|---|
| ConvNext-Tiny [29] | 22.7 |
| + diffusion loss TTA | 19.8 **(-2.9)** |
| + timestep aug | 24.5 **(+4.8)** |
| + noise aug | 24.7 **(+0.2)** |
| + adapting diffusion weights | **25.8 (+1.1)** |

Table 5: **Ablative analysis** of Diffusion-TTA components for ConvNext classifier and DiT diffusion model. We evaluate single-sample adptation for classification on ImageNet-A.

|  | ImageNet | ImageNet-R |
|---|---|---|
| ResNet18 | 68.4 | 37.0 |
| random init | 0.7 | 4.0 |
| adapt logits | 71.8 | 31.0 |
| adapt logits + ensemble | 72.6 | 36.5 |
| adapt BN | 69.0 | 37.0 |
| adapt last FC layer | 68.6 | 37.5 |
| adapt classifier | 73.0 | 38.0 |
| adapt classifier + adapt diffusion (Ours) | **78.2** | **42.6** |

Table 6: **Ablative analysis** of model / layer adaptation. We evaluate single-sample adaptation for classification on ImageNet and ImageNet-R. Our Diffusion-TTA adapts both the classifier (ResNet18) and diffusion model (DiT), achieving best adaptation improvement.

## 4.5 Discussion - Limitations

Diffusion-TTA optimizes pre-trained discriminative and generative models using a generative diffusion loss. A trivial solution to this optimization could be the discriminative model $f_\theta$ overfitting

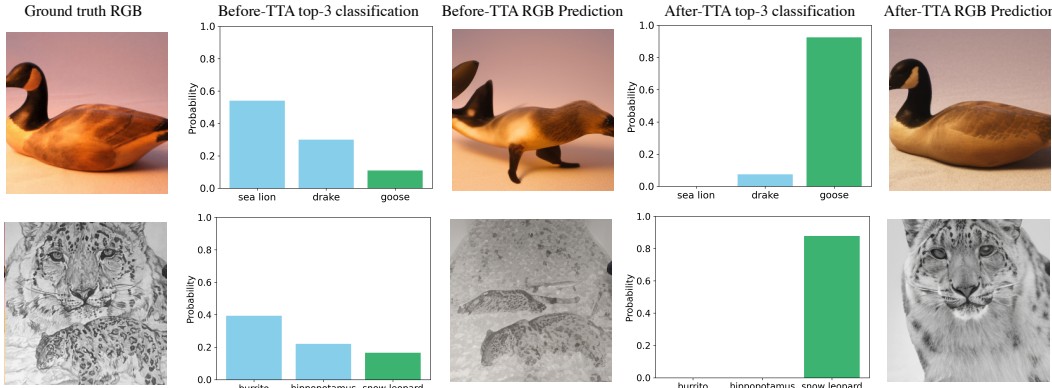

Figure 5: **Visualizing Diffusion-TTA improvement across adaptation steps.** From left to right: We show input image, predicted class probabilities **before** adaptation (green bars indicate the ground truth category), predicted RGB image via DDIM inversion when the diffusion model is conditioned on predicted class probabilities, the predicted class probabilities **after** adaptation and predicted RGB image after adaptation. As can be seen, Classification ability of the classifier improves as diffusion model's ability to reconstruct the input image improves, thus indicating that there is strong correlation between the diffusion loss and the classification accuracy.

to the generative loss and thus predicting precisely $p_\phi(\mathbf{c}|x)$. Empirically we find that this does not happen: Diffusion-TTA consistently outperforms Diffusion Classifier [27], as shown in Table 7. We conjecture that the reason why the discriminative model does not overfit to the generative loss is because of the weight initialization of the (pre-trained) discriminative model, which prevents it from converging to this trivial solution. For instance, if the predictions of the generative and discriminative model are too far from one another that is: distance$(p_\phi(\mathbf{c}|x), p_\theta(\mathbf{c}|x))$ is too high, then it becomes very difficult to optimize $\theta$ to reach a location such that it predicts $p_\phi(\mathbf{c}|x)$. To verify our conjecture, we run a few ablations in Table 6, i) We randomly initialize $\theta$, ii) Instead of optimizing $\theta$ we directly optimize the conditioning variable $\mathbf{c}$. In both cases, we find that results are significantly worse than when optimizing the pre-trained discriminative model.

Diffusion-TTA modulates the task conditioning of an image diffusion model with the output of the discriminative model. While this allows us to use readily available pre-trained models it also prevents us from having a tighter integration between the generative and the discriminative parts. Exploring this integration is a direct avenue of future work. Currently the generative model learns $p(x|y)$, where $x$ is the input image. Training different conditional generative model for different layers such as $p(l1|y)$ or $p(l1|l2)$, where $l1$ and $l2$ are features at different layers, could provide diverse learning signals and it is worth studying. Finally, Diffusion-TTA is significantly slow as we have to sample multiple timesteps to get a good estimate of the diffusion loss. Using Consistency Models [43] to improve the test-time-adaptation speed is a direct avenue for future work.

## 5 Conclusion

We introduced Diffusion-TTA, a test time adaptation method that uses generative feedback from a pre-trained diffusion model to improve pre-trained discriminative models, such as classifiers, segmenters and depth predictors. Adaptation is carried out for each example in the test-set by backpropagating diffusion likelihood gradients to the discriminative model weights. We show that our model outperforms previous state-of-the-art TTA methods, and that it is effective across multiple discriminative and generative diffusion model variants. The hypothesis of visual perception as inversion of a generative model has been pursued from early years in the field [33, 53].The formidable performance of today's generative models and the presence of feed-forward and top-down pathways in the visual cortex [15] urges us to revisit this paradigm. We hope that our work stimulates further research efforts in this direction.

**Acknowledgements**    We thank Shagun Uppal, Yulong Li and Shivam Duggal for helpful discussions. This work is supported by DARPA Machine Common Sense, an NSF CAREER award, an AFOSR Young Investigator Award, and ONR MURI N00014-22-1-2773. AL is supported by the NSF GRFP DGE1745016 and DGE2140739.

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

# A   Appendix

We present Diffusion-TTA, a test-time adaptation approach that modulates the text conditioning of a text conditional pre-trained image diffusion model to adapt pre-trained image classifiers, large-scale CLIP models, image pixel labellers, and image depth predictors to individual unlabelled images. We show improvements on multiple datasets including ImageNet and its out-of-distribution variants (C, R, A, V2, and S), CIFAR-100, Food101, FGVC, Oxford Pets, and Flowers102 over the initially employed classifiers. We also show performance gain on ADE20K and NYU Depth v2 under various distribution drifts over the pre-trained pixel labellers and depth predictors. In the Supplementary, we include further details on our work:

1. We provide comparison to Diffusion Classifier on open-Vocabulary classification in Section A.1.
2. We provide additional results with DiT on ObjectNet dataset in Section A.2.
3. We present architecture diagrams for the tasks of image classification and semantic segmentation in Section A.3
4. We provide a detailed analysis of the computational speed of Diffusion-TTA in Section A.4.
5. We detail the hyperparameters and input pre-processing in Section A.5.
6. We provide details of the datasets used for our experiments in Section A.6.

|  | Food101 | CIFAR-100 | FGVC | Oxford Pets | Flowers102 | ImageNet |
|---|---|---|---|---|---|---|
| Diffusion Classifier [27] | 77.9 | 42.7 | 24.3 | 85.7 | 56.8 | 58.4 |
| CLIP-ViT-L/14 | **93.1** | 79.6 | 32.0 | 91.9 | 78.8 | 70.0 |
| + Diffusion-TTA (Ours) | **93.1** (0.0) | **80.6** (+1.0) | **33.4** (+1.4) | **92.3** (+0.4) | **79.2** (+0.4) | **71.2** (+1.2) |

Table 7: **Comparison to Diffusion Classifier on open-vocabulary classification.** Our model outperforms Diffusion Classifier consistently over all benchmark datasets.

## A.1   Comparison to Diffusion Classifier on Open-Vocabulary Classification

We perform single-sample test-tiem adaptation of CLIP classifiers and compare to Diffusion Classifier [27] on open-vocabulary classification. We test on Food101, CIFAR-100, FGVC, Oxford Pets, Flowers102, and ImageNet datasets. As shown in Table 7, Diffusion-TTA consistent improves CLIP-ViT-L/14 and outperforms Diffusion Classifier over all benchmark datasets.

|  | ObjectNet |
|---|---|
| Customized ViT-L/16 classifier [12] | 37.8 |
| + TTT-MAE | 38.9 |
| ResNet18 | 23.3 |
| + Diffusion-TTA | **26.0** (+2.7) |
| ViT-B/32 | 26.6 |
| + Diffusion-TTA | **31.3** (+4.7) |
| ConvNext-Tiny | 41.0 |
| + Diffusion-TTA | **43.7** (+2.7) |

Table 8: **Single-sample test-time adaptation on ObjectNet.** We use ImageNet pre-trained ResNet18, ViT-B/32, and ConvNext as our classifiers. Our method improves pre-trained image classifiers on out-of-distribution images of ObjectNet.

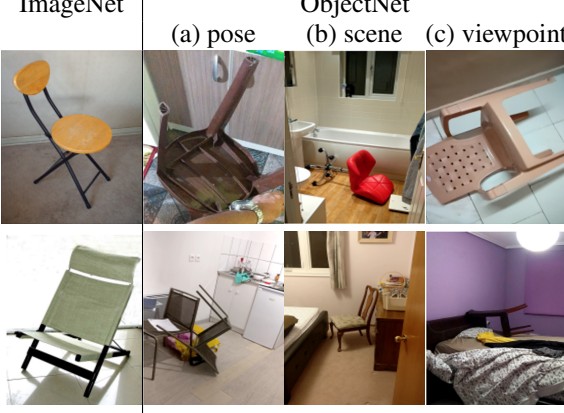

Figure 6: **Comparison of image samples from ImageNet and ObjectNet.** ObjectNet images have more (a) different poses, (b) complex background scenes, and (c) camera viewpoints.

## A.2 Test-time Adaptation of Pre-trained ImageNet Classifiers on ObjectNet

We extend Table 1 in our paper to ObjectNet [4] dataset. ObjectNet is a variant of ImageNet, created by keeping ImageNet objects in uncommon configurations and poses as shown in Figure 6. ObjectNet has 113 common classes with ImageNet, a single class in ObjectNet can correspond to multiple classes in ImageNet. If the predicted class belongs in it's respective set of ImageNet classes, then it's considered as 'correct' if not 'incorrect'. We report the results for ObjectNet in Table 8, following the setup of Section 4.1 in the main paper. Our Diffusion-TTA consistently improves pre-trained ImageNet classifiers with different backbone architectures.

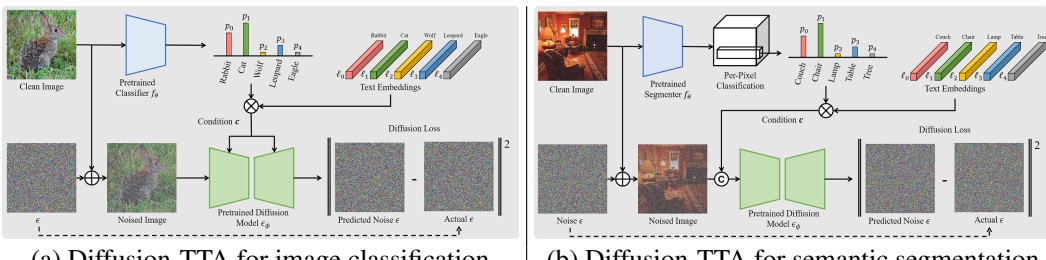



(a) Diffusion-TTA for image classification     (b) Diffusion-TTA for semantic segmentation



Figure 7: **Architecture diagrams for the tasks of (a) image classification and (b) semantic segmentation**. For image classification, we generate a text prompt vector (condition **c**) by doing a weighted average over the probability distribution predicted by the pretrained classifier. We then condition **c** on a pre-trained text-conditioned diffusion model via adaptive layer normalization or cross attentions. For semantic segmentation, we generate per-pixel text prompt (condition **c**) using a pre-trained segmentation model. We then condition **c** on a pre-trained segmentation conditioned Diffusion models via channel-wise concatenation. In the Figure on © denotes channel-wise concatenation.

## A.3 Architecture Diagrams for the Tasks of Image Classification and Semantic Segmentation

We show architectural details for the tasks of image classification and semantic segmentation in Figure 7. Both tasks share the same architectural design, but differ in the way to condition on **c**. For image classification, we generate a text prompt vector (condition **c**) by weighted averaging over the probability distribution predicted by the pre-trained classifier. We condition **c** on a pre-trained text-conditioned diffusion model via adaptive layer normalization or cross attentions. For semantic segmentation, we generate per-pixel text prompt (condition **c**) using a pre-trained image pixel labeller. We condition **c** on a pre-trained segementation conditioned Diffusion models via channel-wise concatenation.

## A.4 Analysis of Computation Speed

We compare the computational latency between Diffusion-TTA and Diffusion Classifier [27] on image classification. Diffusion Classifier [27] is a discrete label search method that requires a forward pass through the diffusion model for each category in the dataset. The computational time increases linearly with the number of categories. In contrast, Diffusion-TTA is a gradient-based optimization method that updates model weights to maximize the image likelihood. The computational speed depends on the number of adaptation steps and the model size, not the number of categories. As shown in Figure 8, out method is less efficient than Diffusion Classifier with fewer categories, but more efficient with more categories. Notably, discrete label search methods, like Diffusion Classifier, do not apply to online adaptation and would be infeasible for dense labeling tasks (searching over pixel labellings is exponential wrt the number of pixels in the image).

**Computation compared to Diffusion Classifier.** Diffusion Classifier [27] inverts a diffusion model by performing discrete optimization over categorical text prompts, instead we obtain continuous gradients to search much more effectively over a pre-trained classifier's parameter space. This design choice makes significant trade-offs in-terms of computation costs. For instance, Diffusion Classifier requires to do a forward pass for each category seperately and thus the computation increases linearly

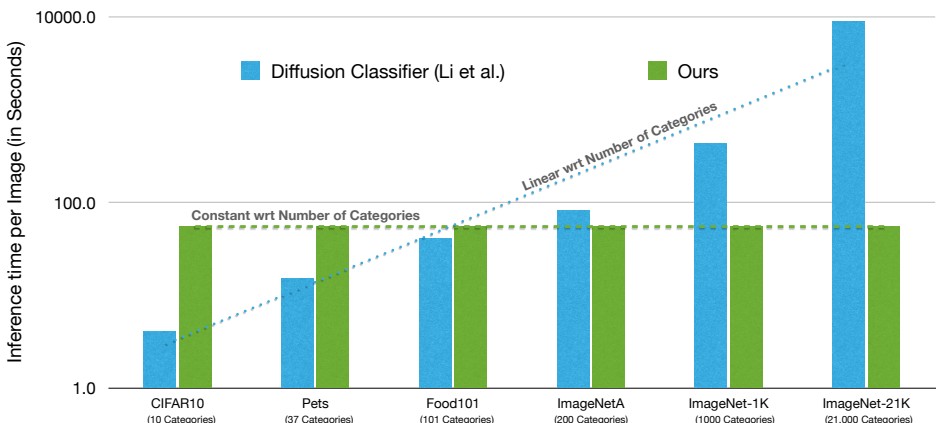

Figure 8: **Comparison of computation speed between Diffusion-TTA and Diffusion Classifier on image classification.** We plot the required computation time with increasing numbers of image categories. Diffusion Classifier [27] requires a forward pass through the diffusion model for each category in the dataset and the computation increases linearly with the number of categories. On the other hand, our method adapts pre-trained classifiers and thus the computation is instead dependent on the throughput of the classifier. Our method becomes more computationally efficient than Diffusion Classifier as the number of categories in the dataset increases.

with the number of categories, however for us it's independent of the number of categories. We compare the per-example inference speed in Figure 8, across various datasets.

## A.5 Hyper-parameters and Input Pre-Processing

For test-time-adaptation of individual images, we randomly sample 180 different pairs of noise $\epsilon$ and timestep $t$ for each adaptation step, composing a mini-batch of size 180. Timestep $t$ is sampled over an uniform distribution from the range 1 to 1000 and epsilon $\epsilon$ is sampled from an unit gaussian. We apply 5 test-time adaptation steps for each input image. We adopt Stochastic Gradient Descent (SGD) (or Adam optimizer [24]), and set learning rate, weight decay and momentum to 0.005 (or 0.00001), 0, and 0.9, respectively. To isolate the contribution of the improvement due to the diffusion model we do not use any form of data augmentation during test-time adaptation.

We use Stable Diffusion v2.0 [40] to adapt CLIP models. For the CLIP classifier, we follow their standard preprocessing step where we resize and center-crop to a resolution of $224 \times 224$ pixels. For Stable Diffusion, we process the images by resizing them to a resolution of $512 \times 512$ pixels which is the standard image size in Stable Diffusion models. For the CLIP text encoder in Stable Diffusion and the classifier we use the text prompt of "a photo of a <class_name>", where <class_name> is the name of the class label as mentioned in the dataset.

For the adaptation of ImageNet classifiers, we use pre-trained Diffusion Transformers (DiT) [36] specifically their XL/2 model of resolution $256 \times 256$, which is trained on ImageNet1K. For ImageNet classifiers we follow the standard data pre-processing pipeline where we first resize the image to $232 \times 232$ and then do center crop to a resolution of $224 \times 224$ pixels. For DiT we resize the image to $256 \times 256$, before passing it as input to their model.

## A.6 Datasets

In the following section, we provide further details on the datasets used in our experiments. We adapt CLIP classifiers to the following datasets incuding ImageNet: **1. CIFAR-100** [26] is a compact, generic image classification dataset featuring 100 unique object classes, each with 100 corresponding test images. The resolution of these images is relatively low, standing at a size of $32 \times 32$ pixels. **2. Food101** [6] is an image dataset for fine-grained food recognition. The dataset annotates 101 food categories and includes 250 test images for each category. **3. FGVC Airplane** [30] is an image dataset for fine-grained categorization containing 100 different aircraft model variants, each with 33 or 34 test images. **4. Oxford Pets** [34] includes 37 pet categories, each with roughly 100 images for

| Food101 | CIFAR-100 | FGVC | Oxford Pets | Flowers102 | ImageNet |
|---------|-----------|------|-------------|------------|----------|

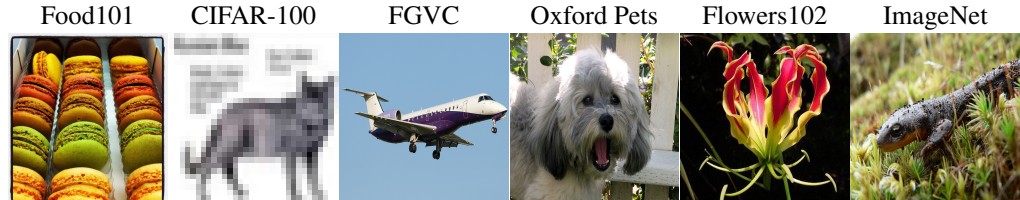

Figure 9: **Image datasets for zero-shot classification.** From left to right: we show an image example obtained from Food101, CIFAR-100, FGVC, Oxford Pets, Flowers102, and ImageNet dataset. CLIP classifiers are not trained but tested on these datasets.

testing. **5. Flower102** [32] hosts 6149 test images, annotated in 102 flower categories commonly found in the United Kingdom. For all of these datasets, we sample 5 images per category for testing.

| ImageNet | ImageNet-C | ImageNet-A | ImageNet-R | ImageNetV2 | ImageNet-S |
|----------|------------|------------|------------|------------|------------|

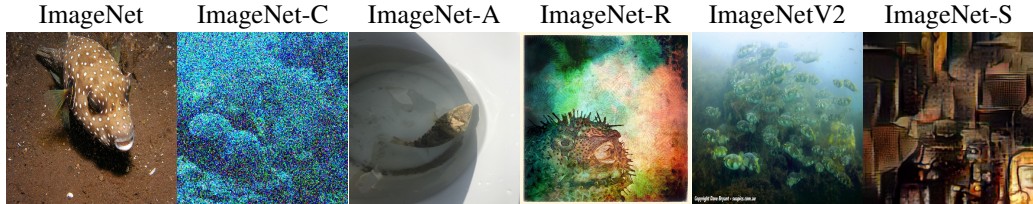

Figure 10: **Image samples from ImageNet and its out-of-distribution variants.** From left to right: we show images of the *puffer* category in the ImageNet, ImageNet-C, ImageNet-A, ImageNet-R, ImageNetV2, and Stylized ImageNet dataset. ImageNet-C applies different corruptions to images. ImageNet-A consists of real-world image examples that are misclassified by existing classifiers. ImageNet-R renders images in artistic styles, e.g. cartoon, sketch, graphic, *etc*. ImageNetV2 attempts to collect images from the same distribution as ImageNet, but still suffers from minor distribution shift. ImageNet-S transforms ImageNet images into different art styles, while preserving the global contents of original images. Though these images all correspond to the same category, they are visually dissimilar and can easily confuse ImageNet-trained classifiers.

We consider ImageNet and its out-of-distribution variants for test-time adaptation. **1. ImageNet** [9] is a large-scale generic object classification dataset, featuring 1000 object categories. We test on the validation set that consists of 50 images for each category. **2. ImageNet-C** [19] applies various visual corruptions to the same validation set as ImageNet. We consider the shift due to gaussian noise (level-5) in our single-sample experiments. We consider 5 types of shift (gaussian noise, fog, pixelate, contrast, and snow) in our online adaptation experiments. **3. ImageNet-A**[21] consists of real-world image examples that are misclassified by existing classifiers. The dataset convers 200 categories in ImageNet and 7450 images for testing. **4. ImageNet-R** [20] is a synthetic dataset that renders 200 ImageNet classes in *art*, *cartoons*, *deviantart*, *graffiti*, *embroidery*, *graphics*, *origami*, *paintings*, *patterns*, *plastic objects*, *plush objects*, *sculptures*, *sketches*, *tattoos*, *toys*, and *video game* styles. The dataset consists of 30K test images. **5. ImageNetV2** collects images from the same distribution as ImageNet, composed of 1000 categories, each with 10 test images. **6. ImageNet-S** transforms ImageNet images into different artistic styles, while preserving the global contents of original images. For ImageNet and its C, R, V2, and S variants except A, we sample 3 images per category, as these datasets contains a lot more classes than the other datasets. For ImageNet-A we evaluate on it's whole test-set.

