# OpenReview forum: "Diffusion-TTA: Test-time Adaptation of Discriminative Models via Generative Feedback"
_NeurIPS.cc/2023/Conference — NeurIPS 2023 poster_

### Official Review · Reviewer_4gPG · 2023-06-19

**Soundness:** 3 good
**Presentation:** 3 good
**Contribution:** 3 good
**Rating:** 7
**Confidence:** 3

**Summary:**

The authors propose combining a diffusion model with a discriminative model for test-time adaptation: They propose adapt a pretrained classifier using a diffusion model with its likelihood loss. They show results on several OOD robustness benchmarks, commonly used in TTA.

**Strengths:**

I like the idea of combining a generative model with a discriminative model to try to get the benefits of both. The proposed method is elegant, but unfortunately heavy on the compute side.

The paper is well-written and easy to follow. The approach is clear.

In general, I think the method is interesting and the idea looks novel to me. But I do think there are several major issues with the paper which need to be resolved. I am happy to engage in a discussion with the authors and will raise my rating if my points are addressed.

**Weaknesses:**

Weaknesses:
My biggest issue is with the experimental results. I find them impossible to judge (see below) and thus, do not understand whether this method works similarly or better than other methods. Given that the proposed method is very heavy on the compute side, I think it would need to make up for it in much stronger results. If e.g. this method performs similarly to TENT / COTTA, then people would still use TENT / COTTA because those methods are very cheap to use. Given that this is an empirical and not a theory paper, I also think more results / ablations should be presented.

### Issues with the results section
Table 2: 1) It is not clear which baseline models were used for TENT / COTTA / TTT here. The final numbers heavily depend on the chosen architecture and the results are only comparable if the same architecture was chosen. Please indicate the architecture for the baseline methods both in the text and in the Figure captions. 2) It is not clear whether accuracy or error are displayed in Table2. Please indicate this, and also please add \uparrow and \downarrow to indicate whether a higher or a lower number is better. An accuracy (if accuracy is displayed?) of 11% on ImageNet-C looks quite bad and I am wondering where this number comes from? The TENT authors report an accuracy of 44% for a ResNet50 in their paper. I think it would be necessary for the authors to add results for a ResNet50 such that the numbers are comparable.

Right now, the results look strange. If we assume that the architecture is the same, then it seems TENT strongly degrades performance. E.g. on ImageNet-R, the TENT number is 24.3 for TENT, 34.6 for the baseline RN18 and 39.7 for Diffusion-TTA. Does this mean that there is an optimization issue because TENT should not degrade performance or is it a different architecture? In that case, the numbers are not comparable. Please rework Table 2 as following: Show the baseline performance without adaptation in the first row, then report numbers from the literature for this architecture, then report your results on the same architecture.


### General comments
Line 36: “In this paper, we take an alternative perspective. Instead of considering generative and discriminative models as competitive, we argue that they should be coupled in a way that leverages the best of both worlds: discriminative models are good at building powerful conditional density models but overfit to training distribution, and generative models generalize better but struggle to learn discriminative features.” -> minor: to the training distribution. The statement that “generative models generalize better” seems unsupported to me. As far as I see, [18] does not provide evidence for better OOD generalization for generative models. Further Carlini et al. [A] show that diffusion models actually memorize their training data and emit the memorized data at test time. Could the authors comment on how they think the findings presented in [A] square with the results in this paper? If diffusion models memorize their training data, I find it unintuitive that they would learn features that would allow them to generalize to OOD data. Put simply, if the task can be solved with memorization, there is no need to learn generalizable features.

[A] Carlini et al. “Extracting Training Data from Diffusion Models”, https://arxiv.org/pdf/2301.13188.pdf

In my opinion, the paper lacks intuition and needs a better explanation why the method works. As written above, it is not evident to me that generative models should have better generalizable features compared to discriminative models. I am wondering whether the approach works because of an effect similar to model ensembling where we know that combining predictions of multiple models leads to better results than using a single model. I wonder if using a diffusion model has a similar effect, because it regularizes the weights to be compatible with both the discriminative and the generative model. Could the authors comment on this thought / better explain why their method works?

Do the authors think one could train a discriminative network from scratch using their approach? If not, why do they think it (only?) helps in the test time adaptation stage? Is it maybe that one finds the “correct” loss basin with regular training and then finds a better loss value within that basin with test time adaptation? I am thinking of the paper by Wortsman et al. [B] where the authors showed that one can interpolate weights of different models because they are in the same low error basin. Could the authors maybe compare the training and test losses (regular cross-entropy) of the baseline and test-adapted models when interpolating the weights from the regular to the adapted model? I am wondering whether the loss goes down monotonically.

 [B] Wortsman et al. “Model soups: averaging weights of multiple fine-tuned models improves accuracy without increasing inference time”, https://arxiv.org/abs/2203.05482


The text in Figure 5 is too small below the bars and largely unreadable.

Figure 6: It would be helpful to repeat this analysis on some other dataset. The solo peak at K=5 somewhat looks like an artifact and I wonder whether one would observe similar behavior in other datasets. How do the authors interpret this strange “peakiness” of the curve at K=5?

More discussion on the ablation analysis would be nice. Why is it that “using a single randomly sampled timestep and noise latent (+diffusion TTA), we find a significant reduction in the classification accuracy (−2.9%)”?

I think the authors should perform a complexity analysis since their method involves making a forward and a backward pass through a large diffusion model to make use of its loss. This is a large overhead compared to much simpler TTA methods such as TENT. While there is a complexity analysis in the Appendix, Fig.2, the overhead is not compared to e.g. TENT which should be much faster.

In a similar vein, I would argue it will be hard to actually find an application for this method since the computational requirements are so massive.  The authors write that “We conduct our experiments on a single NVIDIA-A100 40GB VRAM GPU (…). Since our GPU fits only a batch size of 20, (…)”. TTA methods are usually designed to be light-weight such that they can be applied on the fly, e.g. if an autonomous car needs to adapt to changing weather conditions. This method does not fulfil the requirements usually considered for TTA, and thus, will be of rather limited use for practitioners.

Line 173: “For all experiments, we adjust all the parameters of the classifier.” I am curious about this choice. Several papers show that at least for ResNets, it is better to do TTA with affine BN layers [C, D]. Does this method not work when adapting affine BN layers or does it work better if all layers are adapted?

[C] Wang et al. “Tent: Fully Test-time Adaptation by Entropy Minimization”

[D] Rusak et al. “If your data distribution shifts, use self-learning”

The appendix should be referenced in the main text.


**Questions:**

Please respond / fix the points I outlined above.

Please fix Table 2 to make baselines comparable to your results. Please add a RN50 baseline since it is common to report TTA results on this architecture.

I would advise the authors to do a round of proof-reading of the manuscript, as there are some grammar errors and typos, e.g. line 234, line 123: adaptation


**Limitations:**

The authors have not included a Broader Impact section and did not discuss the potential negative societal impact of their work. I think they could discuss how the chosen diffusion model with affect the adapted classifier. Practitioners should note that biases present in the diffusion model will likely also manifest in the adapted classifier. The classifier itself may or may not have biases of its own which in turn may or may not be corrected by the diffusion model.

One limitation of the method is that it is very compute-intense and thus will likely be of limited use to practitioners.

---

> ### Author Rebuttal · Authors · 2023-08-09
>
> **Q6.1: Backbone architecture unclear for baselines.**
>
> Please see Q1.1 and Q1.2 in the global comment.
>
> **Q6.2: Add RN50 backbone to baselines.**
>
> We have included ResNet50 in our online TTA results in Q1.2.  Here, we report TTA results using ResNet50 under single-example settings.
>
> ||ImageNet|ImageNet-R|ImageNet-V2|
> |:---|:----:|:----:|:----:|
> |**ResNet50**|76.0|40.0|63.5|
> |$~~~~$+CoTTA|70.4 (-5.6)|39.2 (-0.8)|57.6 (-5.9)|
> |$~~~~$+TENT|70.4 (-5.6)|39.3 (-0.7)|57.7 (-5.8)|
> |$~~~~$+Dif-TTA (single-example)|**78.6 (+2.6)**|**42.5 (+2.5)**|**66.9 (+3.4)**|
>
> **Q6.3: TENT accuracy very low, Accuracy or Error not mentioned.**
>
> We report the accuracy in all our Tables. In our online adaptation results (see Q1.2), we show that TENT and CoTTA only improve on certain distribution shifts (ImageNet-C shifts) and classifiers (ResNet). They both fail to improve classification on other ImageNet variants (see Q1.1).  Our conjecture is that TENT and CoTTA are online adaptation methods, and they are not robust when input examples share very little distribution shifts.
>
> A similar conclusion can be found in a recent paper  “On Pitfalls of Test-Time Adaptation”[1]. The authors show that methods that primarily work under online settings (TENT or CoTTA) are not robust to different types of distribution shifts (as reported in Table 4 in their paper).  In fact, these methods turn out to be very sensitive to model architectures and optimization hyper-parameters.
>
> Our method instead exploits a strong generative prior and thus is more robust.
>
> **Q6.4: Unsupported claim of better generalization of generative models. Comment on diffusion models memorize train data.**
>
> Thanks for the suggestion, We will remove the sentence regarding better generalization of generative models  and revise the justification of our method as follows:
>
> Diffusion Classifier[1] and Clark et al [2] explored how to use generative models (diffusion models) for the task of classification. They find that while discriminative models are better at learning unary concepts, generative models offer distinct benefits. In particular, in Table 2 of [2], the authors show that generative models (Imagen) exhibit reduced texture bias than state-of-the-art discriminative models (CLIP-ViT/L-14 or ViT-22B).  In Table 2 of [1], generative models (Stable Diffusion) outperform state-of-the-art discriminative models (CLIP-ViT/L-14 and OpenCLIP-ViT/H-14)  in modeling object relations on Winoground dataset. Motivated by the desire to leverage the strengths of discriminative models and generative models, we propose to combine them using test-time adaptation.
>
> [1] Your diffusion model is secretly a zero-shot classifier. Li et al.
>
> [2] Text-to-image diffusion models are zero-shot classifiers. Clark, K et al.
>
> **Q6.5: Paper lacks intuition and needs a better explanation why it works?**
>
> We conjecture that Diff-TTA works, as both discriminative and generative models capture distinctive aspects of the data, as discussed in previous question.
>
> Our method integrates both these models. Specifically
> - It optimizes the classifier to achieve high image likelihood under the loss landscape of a pre-trained diffusion model
> - It optimizes the diffusion model to have image likelihood given the output of a pre-trained classifier.
>
> We think in satisfying both these constraints, it optimizes for a form of consensus between the two models.
>
> To better understand what’s happening, we compare Diff-TTA against the following baselines.
>
> - Logit-Adapt: In this method we do not use any pre-trained classifier, instead we initialize the logits using zeros and optimize them per example using diffusion objective.
>
> - Logit-Adapt-Ensemble: We average the probabilities of the above Logit-Adapt baseline with the probabilities predicted by the pre-trained classifier.
>
> - Classifier-Adapt: In this method we adapt the pre-trained classifier weights while freezing the diffusion model.
>
> - Diff-Adapt + Classifier-Adapt (Ours): In this method we adapt both the pre-trained classifier weights and the diffusion weights. This refers to our method Diff-TTA.
>
> ||ResNet18|Logit-Adapt|Logit-Adapt-Ensemble|Classifier-Adapt|Diffusion-Adapt + Classifier-Adapt (Ours)|
> |:---|:----:|:----:|:----:|:----:|:----:|
> |ImageNet|68.4|71.8|72.6|73.0|**78.2**|
> |ImageNet-R|37.0|37.0|36.5|38.0|**42.6**|
>
> **Q6.6: Can a discriminative model be trained from scratch?**
>
> We find that training a classifier scratch from does not work. Our conjecture is that good initialization of the classifier makes the loss landscape more convex and therefore easier to optimize.
>
> ||Adapt classifier from scratch |Adapt  pre-trained classifier|
> |:---|:----:|:----:|
> |ImageNet|0.7 |**78.2**|
> |ImageNet-R|4|**42.5**|
>
> **Q6.7: Visualize loss curves vs interpolating weight constant, akin to Model Soup?**
>
> Thanks for this suggestion.  Please refer to Figure 2 in the attached pdf, where we make this plot. We indeed find that, in instances where Diff-TTA correctly classifies the image, the loss monotonically reduces as we interpolate between the model weights.
>
> **Q6.7: K evaluation on more datasets**
>
> Please see our response to Q2.3.
>
> **Q6.8: Why single sample timestep doesn't work**
>
> Our method performs TTA by minimizing the variational lower bound. Using a single timestep instead of an expectation over multiple timesteps results in a bad approximation of the lower bound, thus hurting performance. In Figure 3 of the attached pdf, we visualize this phenomenon using a test sample.
>
> **Q6.9: Heavy compute cost.**
>
> Please see our response to Q3.2.
>
> **Q6.10: Complexity analysis compared to TENT**
>
> Using a single or multiple gpu, Diff-TTA takes 55 or 1.2 seconds to adapt on one image, though TENT takes 0.03 sec per image.
>
> **Q6.11: Adapting just the BN Layers vs all layers**
>
> Please see our response to Q4.4.
>
> **Q6.12: Discuss Negative Impact**
>
> Thanks for the bringing this up. Due to lack of space we will include this in our final paper.

---

> > ### Comment · Reviewer_4gPG · 2023-08-12
> > **Response to authors' rebuttal**
> >
> > Dear authors,
> >
> > thank you for addressing my concerns in your rebuttal. I think you did a great job providing new results! In particular:
> >
> > - I appreciate reporting standardized results for your method vs TENT / CoTTA. It is nice to see that your method works best.
> > - Thank you for adding semantic segmentation results.
> > - I think your response to Q.6.7 is really interesting and Figure 2 in the attached pdf is insightful. I think it helps understanding the method better and would encourage you to include the results in the main manuscript, if space permits.
> > - It is also interesting that the model cannot be trained from scratch using the method, but that it only works for finetuning. Again, I think this provides us with insight about the method and should be included in the final version.
> >
> > One concern remains and that is of the computational complexity of the method. This method is 2-3 orders of magnitude slower than TENT, but then TENT does not always work and also cannot be applied in the single-sample regime. Given that TTA methods typically need to be light-weight, I am not sure how widely this method will be used in practice. I don't know how big of a concern this is, as this paper can be regarded as a "proof-of-principle" contribution. Could the authors maybe try to elaborate in which settings they think this method could be used successfully?
> >
> > I have raised my score because I think the authors did a great job addressing my concerns.
> >
> > Best,
> > Reviewer 4gPG

---

> > > ### Author Response · Authors · 2023-08-12
> > >
> > > We would like to sincerely thank the reviewer for all their insightful comments. They really help us improve our work! As recommended, we will incorporate the results and ablations in the paper's final version.
> > >
> > > We believe our current method holds promise for offline test-time adaptation. In scenarios where static, unsupervised data from a new domain is available, our method can adeptly adjust to distribution shifts. Unlike online methods, offline TTA approaches do not necessitate being lightweight.
> > >
> > > However, we also recognize the potential of methods like Consistency models [1]. Such models can strike a balance between computational efficiency and accuracy, potentially optimizing computation speed in a significant way.
> > >
> > > [1] Consistency Models. Song et al. ICML 2023.

---

### Official Review · Reviewer_VyeN · 2023-07-02

**Soundness:** 2 fair
**Presentation:** 2 fair
**Contribution:** 3 good
**Rating:** 6
**Confidence:** 4

**Summary:**

The paper discusses a method that uses generative models as test-time adapters for discriminative models. The authors propose a technique to adapt pre-trained classifiers and CLIP models to individual unlabeled images. They achieve this by modifying the text conditioning of a text-conditional pretrained image diffusion model and maximizing the image likelihood through backpropagation. The proposed approach improves classification accuracy on various datasets, including ImageNet. The authors compare their method with previous test-time adaptation techniques and demonstrate its superior performance. They highlight that this is the first work to adapt large-scale pre-trained discriminative models to individual images without requiring joint discriminative and self-supervised training objectives.

**Strengths:**

* The paper addresses a fundamental challenge in deep learning models, which is test-time out-of-distribution generalization. This problem is known to be difficult to solve.

* The authors propose an interesting approach that combines generative and discriminative models to tackle this issue. Particularly intriguing is their idea of using a diffusion model as a guide for the classifier, by modulating the text conditioning of the model.

* The experimental evaluation conducted in the paper is good, demonstrating the effectiveness of the proposed method. The authors thoroughly evaluate their approach on various datasets, including conducting ablations to analyze the discriminative and generative components.

Overall, the paper provides a valuable contribution to the field with its innovative approach and robust experimental evaluation.

**Weaknesses:**

* While I agree with your premises regarding the class conditional model, I noticed that the improvements achieved are marginal on certain datasets, in particular for open-set problems. It would be beneficial to have a more in-depth analysis specifically focusing on ImageNet for open-set evaluation.

* Additionally, it would be valuable to incorporate confidence intervals in the evaluation to provide a better understanding of the statistical significance of the results. Furthermore, I believe it would be advantageous to include additional metrics such as top-k accuracy and f-score (where possible), rather than solely relying on accuracy. These metrics would provide a more comprehensive evaluation of the model's performance in various scenarios.

* It is worth mentioning that CLIP, even without any adaptation, seems to perform well, and the extent of improvement achieved by the proposed method is not decisively evident in most cases, especially without the inclusion of confidence intervals.

* Finally, the clarity of the introduction can be improved. I think a better discussion of discriminative vs generative models would be a valuable addition to this paper.

**Questions:**

* How are the learned class or text embeddings obtained, specifically referring to line 156 or Fig.1? Are the embeddings the same as those employed by the classifier or CLIP models? If so, does this imply that the class/text embeddings change during test time adaptation?

* What is the time required to adapt the model to a single image? Can this approach be scaled up for practical adaptation applications?

* In Figure 6, have you investigated the effects of increasing the value of K for a CLIP model?

**Limitations:**

The obtained results are interesting; however, it is worth noting that the observed improvements, particularly in relation to the CLIP models, appear relatively small. It is interesting that the base classifiers, in particular the CLIP models, already exhibit good performance even without any fine-tuning or test-time adaptation.

---

> ### Author Rebuttal · Authors · 2023-08-10
>
> **Q5.1: Marginal Improvements in Open-Set Datasets**
>
> You are correct that the improvements on open-set problems are not as high on the closed-set problems. We think this is mainly because we use Stable Diffusion as our diffusion model.  Stable Diffusion conditions the image generation on the text embeddings from CLIP . However, the language model in CLIP is much weaker than open ended large-language models like T5-XXL which is used in Imagen. Clark etal [1] validates this claim, where they find Imagen results in a significantly better classifier than Stable Diffusion. We don’t have this issue for DiT as it learns its own text embedding for each class.
>
> Here, we compare our method against a strong baseline–TPT [2]. TPT is a test-time adaptation method for large-scale text-to-image models such as CLIP. TPT optimizes the text prompts to encourage consistent predictions across augmented views of the same test image by minimizing the marginal entropy. Our method on the other hand optimizes the network parameters using generative feedback.
>
> Please see our response to Q2.4 for the comparison among our method against a strong baseline–TPT.  We find that Diff-TTA consistently outperforms TPT across all CLIP backbone architectures.
>
> [1] Text-to-image diffusion models are zero-shot classifiers. Clark, K et al. arXiv:2303.15233.
>
> [2] Test-Time Prompt Tuning for Zero-Shot Generalization in Vision-Language Models, NeurIPS 2022.
>
> **Q5.2: ImageNet Variant on Open vocab setting**
>
> Great suggestion! Our Diffusion-TTA improves CLIP-ViT/B-32 on ImageNet variants (ImageNet, ImageNetV2, and ImageNet-R) consistently.
>
> |  | ImageNet | ImageNet-V2 | ImageNet-R |
> | :---      |   :----:   |  :----:   |  :----:   |
> | **CLIP-ViT/B-32** | 56.3  | 51.3 | 55.5 |
> | $~~~~$+Diffusion-TTA (single-example) | **58.1 (+1.8)** | **52.5 (+1.2)** | **58.2 (+2.7)** |
>
> **Q5.3: Confidence intervals/ Top-k to get better understanding of results**
>
> We report confidence intervals of top-1 and top-3 accuracy over 5 random seeds on the ImageNet-R dataset.  We conduct experiments using the ResNet50 classifier and DiT diffusion model. Our performance gains are consistent across different random seeds.  We will include more detailed analysis in our final submission.
>
> |   | Before-TTA | Seed 1 | Seed 2 | Seed 3 | Seed 4 | Seed 5 | Mean |
> | :---      |   :----:   |  :----:   |   :----:   |  :----:   |   :----:   |  :----:   |  :----:   |
> | Top-1 Acc.  | 40.0 | 42.5 | 42.2 | 42.7 | 41.8 | 42.8 |  42.4 ($\pm$ 0.41) |
> | Top-3 Acc.  | 51.7 | 52.0 | 51.8 | 52.2 | 51.8 | 51.8 |  51.9 ($\pm$ 0.13) |
>
>
> **Q5.4: Better discussion of the  differences between generative/discriminative models in the introduction**
>
> Thank you for your suggestion. We will include the following sentences in our intro.
> So far discriminative models have mainly been used for discriminative tasks such as image classification. Recently,  Diffusion Classifier[1] and Clark etal l2], explored the use of recent generative models (diffusion models) for the task of classification. They find that while discriminative models are better at learning unary concepts, generative models offer distinct benefits. Notably, Clark etal in Table 2 of their paper find that generative models such as Imagen exhibit reduced texture bias than state-of-the-art discriminative models such as CLIP-L14 or ViT-22B.  Further Diffusion Classifier[1] in their Table 2 find that generative models such as Stable Diffusion outperform state-of-the-art discriminative models such as CLIP-L14 and OpenCLIP-H14  in modeling object relations on Winoground daetaset. Motivated by the desire to leverage the strengths of discriminative models and generative models, we propose to combine them using test-time adaptation.
>
> The exact form of their combination is  motivated by neuroscience findings that suggest that the brain performs a form of analysis by synthesis by rendering  inferred concepts in an iterative feedback loops of encoding and decoding processes [3]. Our model is a form of analysis by synthesis using conditional diffusion models as the renderer (decoder) and classifiers or segmentors as the encoder.
>
> [1] Your diffusion model is secretly a zero-shot classifier. Li et al. arXiv 2023.
>
> [2] Text-to-image diffusion models are zero-shot classifiers. Clark, K et al. arXiv:2303.15233.
>
> [3] Generative Feedback Explains Distinct Brain Activity Codes for Seen and Mental Images. Breedlove et al.  Current Biology 2020.
>
> **Q5.5: How are text embeddings obtained/updated**
>
> The class or text embeddings are obtained from the pre-trained diffusion model.
>
> - For stable diffusion, the text-embeddings are obtained by encoding each class name in the dataset using  the CLIP text encoder.
> - DiT, on the other hand learns a set of class embeddings for ImageNet classes, we simply use that in our experiments.
> - In our experiments, as we update the diffusion model we do update the text embeddings during TTA.
>
> **Q5.6: Time to Adapt to a Single image?**
>
> In a single GPU, our method takes 55 seconds for adaptation mainly due to sequential gradient accumulation steps.  We can reduce the computation latency to 1.2 seconds per example using multiple gpus.  Also, we can further speed up our method by:
>
> 1. Dynamically deciding when to adapt the model based on the entropy in the diffusion loss.
> 2. Incorporating better timestep weighting mechanisms [1] and recent advances with Consistency Models [2] which allow one-step image generation can provide significant boosts in the speed.
>
> [1] Consistency Models. Song et al. ICML 2023.
>
> [2] Text-to-image diffusion models are zero-shot classifiers. Clark, K et al. arXiv:2303.15233.
>
> **Q5.7: Increasing the value of K for CLIP**
>
> Please see our response to Q2.3

---

> > ### Comment · Reviewer_VyeN · 2023-08-13
> >
> > Thank you for the rebuttal. It answers my questions and I appreciate the new results.

---

> > > ### Author Response · Authors · 2023-08-13
> > >
> > > We also want to thank the reviewer for the questions. They help us improve our work and made us focus more on the ablations.

---

### Official Review · Reviewer_qddk · 2023-07-04

**Soundness:** 3 good
**Presentation:** 4 excellent
**Contribution:** 3 good
**Rating:** 7
**Confidence:** 4

**Summary:**

This paper proposes a test-time adaptation (TTA) method that adjusts pretrained image classifiers at test time by leveraging the power of pretrained, text-conditioned generative diffusion models. Given an unseen image, the proposed method Diffusion-TTA minimizes the diffusion loss with respect to the weights of the pretrained classifier and/or diffusion model via gradient descent at test time. In other words, Diffusion-TTA provides guidance to the classifier to adapt its performance particularly under the out-of-distribution setting.

From the empirical evaluation, Diffusion-TTA shows its effectiveness in encouraging consistent improvements over the initially employed classifier. Furthermore, the method is applicable to utilize different neural net architectures for the classifiers such as ResNet, ViT, and ConvNext-Tiny.

----
After reading the rebuttal, I've raised my score from 6: weak accept to 7: accept.

**Strengths:**

Sufficient novelty:
Diffusion-TTA follows a similar approach from Diffusion Classifier (Li et al. 2023) by utilizing the same objective, i.e., minimizing the diffusion loss, but executing a different optimization approach towards the objective. Diffusion-TTA employs pretrained classifiers and updates the weights via back-propagation of the diffusion loss, while Diffusion Classifier performs a discrete optimization over categorical text prompts to the generative model without using any pretrained classifier. This approach undertaken by Diffusion-TTA, which results in a positive empirical outcome in the context of TTA, is a new perspective to me.

Rigorous ablation study:
The impact of each implementation component defining Diffusion-TTA is empirically investigated and reported so that the readers are aware about the contribution from each component.


**Weaknesses:**

Empirical evaluation:
I think that the added values of Diffusion-TTA over the existing SOTA methods remain a bit inconclusive. Specifically, it’s unclear to me that Diffusion-TTA is generally better than TTA-MAE given two occasions in ImageNet experiments (in-distribution and ImageNet-V2) where TTA-MAE seems to provide slightly or statistically insignificant negative outcomes. On the other hand, TTA-MAE produces larger improvements than Diffusion-TTA.

I’d suggest having comparisons in which the other existing methods use the same backbone networks as Diffusion-TTA for the classifiers (ResNet18, ViT-B/32, or ConvNext-Tiny) whenever applicable.


**Questions:**

How’s the performance of TTA-MAE or other existing TTA methods on the Open-Vocabulary CLIP experiment?

Have the authors tried partial weight updating on the classifier, e.g., only adjusting weights of a few top layers, and observing the performance gain?

In L175, it’s mentioned that adapting the weights of both generator and classifier slightly boosts the performance of Diffusion-TTA for the ImageNet distribution shift case. By how much boost in comparison to keeping the weights of the generator fixed?

**Limitations:**

The limitation in terms of the generality of the method, i.e., only shown to be effective for image classification thus far, has been addressed. Furthermore, although the inference speed is perhaps not a focus in this study, I’d recommend also addressing it (~55 seconds per example on a single NVIDIA-A100 40GB) as another limitation.

I’m hoping that the code implementation is made available soon.

---

> ### Author Rebuttal · Authors · 2023-08-10
>
> **Q4.1:  Improvements over TTT-MAE unclear**
>
> If we consider the **after-TTA absolute scores** our method significantly outperforms TTT-MAE on all except ImageNet-V2 dataset. On ImageNet, ImageNet-A, ImageNet-R, ImageNet-C, and ObjectNet, our method gets (+1.1, +4.5,  +10.5,  +13.9, +4.8)  boosts respectively over TTT-MAE.
>
> We don’t think relative improvement is the right number to track, rather one should track **the absolute score after TTA**. The reason for this is the following: One can easily abuse the relative TTA improvement metric by getting very low before-TTA results, and then showing high gains in performance on top of it, while still having very low absolute after-TTA numbers! On a separate note, adding a different classifier in TTT-MAE  requires re-training the whole network on ImageNet-1K, while for us we can easily plug-in and improve with any pre-trained classifier.
>
> For online TTA, Diffusion-TTA significantly outperforms TTT-MAE across different types of distribution shifts using a smaller classifier: ConvNext (28.5M parameters) vs. customized ViT-L+ViT-B (392M parameters).  Our method achieves 47.4, 65.9 69, 62.4, and 46.2 on Gaussian-Noise, Fog, Pixelate, Snow, and Contrast split in ImageNet-C, compared to 37.9, 51.1, 65.7, 56.5, and 10 obtained by TTT-MAE.  See the online adaptation results in the global comment of Q1.2 for more details.
>
> **Q4.2: Mention the backbones used for COTTA, TENT and TTT-MAE baselines.**
>
> Please see Q1.1 and Q1.2 in the global comment.
>
> **Q4.3: Add TTA baselines on the Open-Vocabulary CLIP experiment.**
>
> We compare our method against TPT [1] on the task of open-vocabulary classification in the following table. TPT is a test-time adaptation method for large-scale text-to-image models such as CLIP.  TPT optimizes the text prompts to encourage consistent predictions across augmented views of the same test image by minimizing the marginal entropy. Our method on the other hand optimizes the network parameters using generative feedback.
>
> [1] Test-Time Prompt Tuning for Zero-Shot Generalization in Vision-Language Models, NeurIPS 2022.
>
>
> | | Food101 | CIFAR100 | FGVC | Pets | Flower102 | ImageNet | Average Improvement |
> | :---      |   :----:   |  :----:   |  :----:   |  :----:   |  :----:   |   :----:   |     :----:   |
> | **CLIP-ViT-B/32** | 78.4 | 60.0 | 18.8 | 77.8 | 64.1 | 56.3 ||
> | $~~~~$+TPT | 79.8 (+1.4) | **65.2 (+5.2)** | 18.4 (-0.4) | 77.3 (-0.5) | 62.8 (-1.4) | 57.5 (+1.2) | +0.91 |
> | $~~~~$+Diffusion-TTA | **80.2 (+1.8)** | 61.8 (+1.8) | **22.2 (+3.4)** | **81.1 (+3.3)** | **64.3 (+0.2)** | **58.1 (+1.8)** | **+2.01** |
> | **CLIP-ViT-B/16** | 84.7 | 68.8 | 21.4 | 80.5 | 67.6 | 60.6 | |
> | $~~~~$+TPT | 85.2 (+0.5) | **68.4 (-0.4)** | 20.4 (-1.0) | 80.0 (-0.5) | **70.0 (+2.4**) | **61.6 (+1.0)** | +0.31 |
> | $~~~~$+Diffusion-TTA | **85.5 (+0.8)** | 67.6 (-0.8) | **22.6 (+1.2)** | **80.8 (+0.3)** | 69.2 (+1.6) | 61.5 (+0.9) | **+0.67** |
> | **CLIP-ViT-L/14** | 91.2 | 79.6 | 29.0 | 89.2 | 75.2 | 68.9 | |
> | $~~~~$+TPT | 90.6 (-0.6) | 80.6 (+1.0) | 30.2 (+1.2) | 87.6 (-1.6) | **76.9 (+1.6)** | **70.6 (+1.7)** | +0.55 |
> | $~~~~$+Diffusion-TTA | **91.2 (0.0)** | 80.6 (+1.0) | **30.6 (+1.6)** | **89.8 (+0.6)** | 76.1 (+0.9) | 69.9 (+1.0) | **+0.85** |
>
>
> We conclude that:
>
> 1. Our Diffusion-TTA consistently outperforms TPT  across all CLIP backbone architectures.
> 2. TPT only adapts prompts and is thus restricted to vision-language classifiers like CLIP.  Our method instead can improve any pre-trained classifier, and even dense pixel labeller (e.g., semantic segmentation)
> 3. Our method works for both single-example and online test-time adaptation, whereas TPT only works on single-example settings.
>
> **Q4.4: Adjusting certain parts of the weights of the classifier?**
>
> We present the ablation study of adapting different layers of the classifier:
>
> 1. Batch Normalization layers only
> 2. The last FC layer only
> 3. The whole classifier
>
> We employ ConvNext-Tiny and subsample 1 image per category for the ablation study. We find that adapting the whole classifier results in the best performance. We will include these ablations in our paper.
>
> | Adapt Classifier: | BN | Last FC layer | Whole model|
> | :---      |   :----:   |  :----:   |  :----:   |
> | ImageNet | 69  |  68.6 | **78.2** |
> | ImageNet-R | 37 |  37.5 | **42.5** |
>
> **Q4.5: Adding an ablation for freezing or not the weights of the image diffusion model**
>
> We present the ablation study of freezing the diffusion model or not.  We employ ConvNext-Tiny and subsample 1 image per category for the ablation study
>
> | Adapt  | Diffusion + Classifier | Classifier only |
> | :---      |   :----:   |  :----:   |
> | ImageNet | **78.2** | 72.6 |
> | ImageNet-R | **42.5** | 37 |
>
> We show that adapting the diffusion model results in significant performance gain. See the same observation in Table 3 in our paper. We will include these ablations in our paper.
>
> **Q4.6: Method is limited to classification.**
>
> Please see Q1.3 in the global comment.
>
> **Q4.7: The proposed method is slow**
>
> In a single GPU, our method takes 55 seconds for adaptation mainly due to sequential gradient accumulation steps.  We can reduce the computation latency to 1.2 seconds per example using multiple gpus.  Also, we can further speed up our method by:
>
> 1. Dynamically deciding when to adapt the model based on the entropy in the diffusion loss.
> 2. Incorporating better timestep weighting mechanisms [1] and recent advances with Consistency Models [2] which allow one-step image generation can provide significant boosts in the speed.
>
> [1] Consistency Models. Song et al. ICML 2023.
>
> [2] Text-to-image diffusion models are zero-shot classifiers. Clark, K et al. arXiv:2303.15233.
>
> **Q4.8: Code Availability?**
>
> Yes, we will open source our code upon paper acceptance.

---

> > ### Comment · Reviewer_qddk · 2023-08-15
> >
> > Thank you for providing a rigorous feedback. I have read all the responses addressed to myself as well as to the other reviewers.
> >
> > I truly appreciate the substantial effort not only clarifying the doubts but also providing new insights and empirical results. I'm now convinced with the effectiveness of the proposed method. I'm happy to promote my score up to +1 level.
> >
> > Please ensure all the new important insights are included in the main manuscript or appendix.

---

> > > ### Author Response · Authors · 2023-08-15
> > >
> > > We would like to thank the reviewer for reading our responses in detail. Their questions about weight ablations and additional baselines majorly improved our work.
> > >
> > > As the reviewer suggested, We will include the new results and insights in the final manuscript.

---

### Official Review · Reviewer_6xj6 · 2023-07-09

**Soundness:** 3 good
**Presentation:** 3 good
**Contribution:** 2 fair
**Rating:** 6
**Confidence:** 4

**Summary:**

Authors aim to improve test-time adaptation of CLIP and ImageNet trained models on different distribution data. They leverage text conditioned image diffusion model to adapt image classifier’s parameters by maximizing image diffusion likelihood. In particular, image classifier’s output probabilities are used as weights for class conditioning the diffusion model and diffusion likelihood gradients are backpropagated to classifier through the probabilities. Adaptation of CLIP across different datasets and adaptation of ImageNet trained models across different ImageNet variants using the diffusion loss show improved top-1 accuracy.

**Strengths:**

1.	Paper is well-written and easy to understand.

2.	Proposed setup is a simple plug and play method that does not require any additional training or warm-up of the models.

3.	Results demonstrate that the proposed adaptation improve the results, particularly benefit smaller models.

4.	Appreciate the analysis that shown diffusion loss and cross-entropy loss are correlated, which support the gains observed in the results.

**Weaknesses:**

1.	Proposed setup can be seen as straight forward extension of Li et al. [22] which introduced similar guidance strategy to optimize over text prompts. Here, classifier parameters are updated instead of text prompts.

2.	Increased time complexity of test time adaptation with the diffusion model, where each sample takes 55 seconds long for adaptation.

3.	Increased model complexity by including diffusion model with large number of parameters in the adaptation process.

4.	The adaptation is not as effective on larger models when compared to smaller models.

5.	Prior works comparisons are included in Table 2 but the network architectures are different across these comparisons and wouldn't be a fair comparison.

**Questions:**

1.	Which diffusion model was used in Sec. 4.1?

2.	CLIP adaptation is shown across multiple datasets. Adapting CLIP with ImageNet variants would be interesting.

**Limitations:**

Yes

---

> ### Author Rebuttal · Authors · 2023-08-09
>
> **Q3.1: Straight forward extension of Diffusion Classifier**
>
> We believe this is not the case for the following reasons:  Diffusion Classifier searches over **discrete** class labels while Diffusion-TTA optimizes the parameter of the diffusion model and the classifier thru gradient descents.  Gradient-based optimization enables us to apply our method to online adaptation settings and extend it beyond classification tasks, such as in the task of semantic segmentation.  In contrast, discrete label search does not apply to online adaptation and would be infeasible for dense labeling tasks (searching over pixel labellings is exponential wrt the number of pixels in the image).
>
> **Q3.2: The proposed method is slow.**
>
> In a single GPU, our method takes 55 seconds for adaptation mainly due to sequential gradient accumulation steps.  We can reduce the computation latency to 1.2 seconds per example using multiple gpus.  Also, we can further speed up our method by:
>
> 1. Dynamically deciding when to adapt the model based on the entropy in the diffusion loss.
> 2. Incorporating better timestep weighting mechanisms [1] and recent advances with Consistency Models [2] which allow one-step image generation can provide significant boosts in the speed.
>
> [1] Consistency Models. Song et al. ICML 2023.
>
> [2] Text-to-image diffusion models are zero-shot classifiers. Clark, K et al. arXiv:2303.15233.
>
>
> **Q3.3: Adaptation improvements smaller on larger models**
>
> That’s indeed true that larger models provide smaller improvements in single example settings. However we find that in online settings we get significant improvements even with larger models.  In the Table below, we show results adapting two of the largest ImageNet Classifiers ViT L/16 (307M parameters) and ConvNext-Large (197M parameters).  We test on the Gaussian-Noise split in ImageNet-C dataset.
>
> | | ConvNext-Large | ViT-L/16 |
> | :---      |   :----:   |  :----:   |
> | No TTA | 37.0 | 34.9|
> | Diffusion-TTA (single-example) | 37.0 | 33.3 |
> | Diffusion-TTA (online) | **54.9 (+17.9)** | **50.0 (+15.1)**|
>
> As can be seen in the table above,  even though our model doesn’t improve in single-example setting, we get significant improvements in the online setting.
>
> **Q3.4: Which diffusion model was used in Sec. 4.1?**
>
> We used Stable Diffusion v1.4 for this section.
>
> **Q3.5: Baselines in Table 2 have different neural architectures**
>
> Please see Q1.1 and Q1.2 in the global comment.
>
> **Q3.6: Show CLIP adaptation on ImageNet dataset variants**
>
> Great suggestion! Our Diffusion-TTA improves CLIP-ViT/B-32 on ImageNet variants (ImageNet, ImageNetV2, and ImageNet-R) consistently.
>
> |  | ImageNet | ImageNet-V2 | ImageNet-R |
> | :---      |   :----:   |  :----:   |  :----:   |
> | **CLIP-ViT/B-32** | 56.3  | 51.3 | 55.5 |
> | $~~~~$+Diffusion-TTA (single-example) | **58.1 (+1.8)** | **52.5 (+1.2)** | **58.2 (+2.7)** |

---

> > ### Author Response · Authors · 2023-08-19
> >
> > **Q3.3: Adaptation improvements smaller on larger models**
> >
> > Dear Reviewer,
> >
> > Following our initial response to your question, we conducted further testing of ConvNext-Large model under various distribution shifts in an online setting. ConvNext-Large is the largest available ConvNext model with 197M parameters, the model achieves state-of-the-art performance on ImageNet.
> >
> > |  | Gaussian-Noise |  &nbsp; &nbsp; Fog | Pixelate | Snow | Contrast |
> > | :---      |   :----:   |    :----:   |  :----:   |  :----:   |  :----:   |
> > | ConvNext-Large | 37.0  | 34.4 | 49.3 | 44.5 | 39.8 |
> > | +   Diff-TTA (online) | **54.9 (+17.9)** | **67.7 (+33.3)** | **71.7 (+22.4)** | **64.8 (+20.3)** | **55.7 (+15.9)** |
> >
> > Based on our results reported above, we find that Diff-TTA consistently achieves significant performance improvements across multiple distribution shifts, reinforcing the adaptability of our method with larger models.
> >
> > Please let us know if you have any additional questions.

---

> > > ### Comment · Reviewer_6xj6 · 2023-08-21
> > > **Response to author's rebuttal**
> > >
> > > I appreciate author's rigorous efforts for preparing the rebuttal. Thanks for conducting the experiments and showing the effectiveness on the larger models. As I'm impressed with the significance of the approach, my original concerns still remain same regarding: a) straight forward extension of Li et al. [22], b) increased time complexity, and c) Increased model complexity. As authors commented that their work is different than Li et al. [22], their argument is not convincing enough. Instead of searching for the class name, the gradients are back-propagated to the classifier in this work, which can be considered a straight-forward extension. As the test time adaptation requires light weight and efficient models, the proposed pipeline is high in both time and memory consumption. However, I also believe that the methodology on diffusion models for test time adaptation is interesting for the community and results are noteworthy. So, I raise my rating by +1.

---

> > > > ### Author Response · Authors · 2023-08-21
> > > >
> > > > Dear Reviewer,
> > > >
> > > > Thank you for your thoughtful feedback and for taking the time to review our rebuttal. We appreciate the constructive criticisms and understand your concerns. We would like to provide further clarity on the distinctions between our work and that of Li et al. [22]:
> > > >
> > > > i) **Task Flexibility**: One of the defining aspects of our approach is its versatility. While Li et al. [22] focuses primarily on Image Classification, our method has a broader scope of applicability. To illustrate, we've showcased results for semantic segmentation in the general comments, but our method is not confined to this; it can also be employed for various tasks, such as depth prediction, pose prediction, and more.
> > > >
> > > > ii) **Adaptation of diffusion weights for added boost**: We believe one of the central contribution of our work lies in the simultaneous adaptation of both the diffusion model weights and the classifier weights. This joint adaptation results in a significant boost in performance.
> > > >
> > > > For example, in an online setting:
> > > >
> > > > - The base classifier of ConvNext yields a performance of **32.3**.
> > > >
> > > > - Adapting only the classifier using the diffusion loss improves the score to **48.9**.
> > > >
> > > > - Crucially, adapting the diffusion weights jointly with the classifier weights improves our performance further to **65.9**.
> > > >
> > > > We've also observed similar performance trends in single-example setting, as shown in Q4.5 of our rebuttal.

---

### Official Review · Reviewer_UVNj · 2023-07-11

**Soundness:** 3 good
**Presentation:** 3 good
**Contribution:** 3 good
**Rating:** 6
**Confidence:** 4

**Summary:**

This work proposed Diffusion-TTA, a test-time adaptation method that, given a test image, updates the classifier’s weights to maximize the image likelihood (i.e., the denoising score matching loss) using the pre-trained diffusion models. Specifically, the classifier uses the clean input image to predict a distribution over labels, which is then used for a weighted sum of learnt class embeddings into a prompt input of diffusion models. In experiments, Diffusion-TTA is tested for adapting both open-vocabulary CLIP models and pretrained image classifiers, where it improves the classification accuracy across different network architectures.

**Strengths:**

1. The idea of using pre-trained diffusion models as image priors to update classifiers at test time is novel. In particular, I like the idea of “using classifier’s output logits for a weighted sum of learnt class embeddings”, which is very intuitive to me.
2. It shows the improvement of classification accuracy across different network architectures after using Diffusion-TTA.
3. The ablation studies (in Table 3) are well conducted to show the importance of each component.

**Weaknesses:**

1. One of my major concerns is about the comparison with baselines in experiments. In Section 4.1 (“adapting open-vocabulary CLIP models”), only Diffusion-TTA works with CLIP, other baselines do not use CLIP. For example, Diffusion Classifier does not use any image classifier, and both Synthetic SD Data and SD Features use the ResNet-50 classifier. I think this setting causes an unfair comparison with baselines. In Section 4.2 (“adapting image classifiers”), I also don’t see what the pretrained image classifiers the baselines (COTTA, TENT and TTT-MAE) have used. It would be good to report their accuracies on the same set of network architectures (ResNet-18, ViT-B/32 and ConvNext-Tiny). In the current version, I’m not sure if the proposed method is better than the baselines under the same conditions.
2. The claim “this is the first work that adapts pre-trained large-scale discriminative models to individual images” is wrong. From what I know, at least TPT [1] is a prior test-time adaptation method using only a single test image. I think TPT is more efficient than the proposed method in adapting CLIP models, since it only updates the input prompts instead of the CLIP weights, and it does not need to backpropagate through large diffusion models with a large batch size.
3. In Figure 6, it looks weird to me that the accuracy curve has a spike at K=5 and does not change after K>=10. Also, a smaller K (maybe K=1?) achieves very similar accuracy with K=5, it seems that we may not need a distribution of output to do a weighted sum. Instead, we can just use the class embedding from the top-1 prediction as the prompt input.
4. Some suggestions regarding the writing presentations: 1) In line 155 and line 156, the variable format is not consistent, such as $z_i$ and $i \in T$. 2) Since the baselines are already discussed in the related work section, there is no need to give a full detailed description of them in experiment section again (line 234-250). They can be moved to Appendix if necessary. 3) The fonts in Figure 5 is too small.
[1] Test-Time Prompt Tuning for Zero-Shot Generalization in Vision-Language Models, NeurIPS 2022.

**Questions:**

1. Can the authors provide results of Synthetic SD Data and SD Features with other image classifiers, such as CLIP-ViT models, to make the comparison more sound? Similarly, can the authors provide results of COTTA, TENT and TTT-MAE across different network architectures (ResNet-18, ViT-B/32 and ConvNext-Tiny)?
2. I suggest the authors add a comparison with TPT in both related work and experiments (adapting CLIP models). How does Diffusion-TTA compare with TPT regarding OOD accuracy and inference efficiency?
3. Can the authors give insights of why the accuracy curve behaves like this (a spike at K=5 and does not change after K>=10) and justify the selection of K (K=5 vs K=1)?

**Limitations:**

The authors have addressed the limitations.

---

> ### Author Rebuttal · Authors · 2023-08-09
>
> **Q2.1: Synthetic SD Data and SD features baseline use ResNet-50 classifier and not CLIP.**
>
> You are correct, Synthetic SD Data, SD features, and Diffusion Classifier are not very relevant or fair baselines for Table 1 of the paper as they don’t use a pre-trained classifier. However, we thought showing their results help readers understand the TTA with discriminative models is critical for recognition tasks.  To avoid confusion, we will remove these baselines from Table 1 and list them in a separate Table in the supplementary.
>
> We followed Diffusion Classifier’s codebase to run Synthetic SD-Data baseline using CLIP.  Notably, SD-Feature baseline is not applicable to CLIP classifier: CLIP takes RGB pixels (HxWx3)  as input, whereas SD-Feature takes high-dimensional latent features (HxWx64) from U-Net as input.
>
>  |  | Food101 | FGVC | Pets |
> | :---      |   :----:   |  :----:   |  :----:   |
> | SD-Data | 12.6 | 9.4 | 31.3 |
> | **CLIP-ViT/B-32** | 78.4 | 18.8 | 77.8 |
> | $~~~~$+SD-Data | 76.5 (-1.9) | 19.4 (+0.6) | 71.2 (-6.6) |
> | $~~~~$+Diffusion-TTA | **80.2 (+1.8)** | **22.2 (+3.4)** | **81.1 (+3.3)** |
>
> We find that SD-Data baseline fails to improve classification on Food101 and Pets dataset (-1.9 and -6.6), while the improvement on FGVC is marginal (+0.6).  Our conjecture is that images generated by Stable Diffusion have a huge distribution shift compared to real images in the test datasets.  Fine-tuning CLIP on synthetic images fails to generalize to real images.
>
> **Q2.2: Mention the backbones used for COTTA, TENT and TTT-MAE baselines.**
>
> Please see Q1.1 and Q1.2 in the global comment.
>
> **Q2.3: Weird Behavior of K  in TopK**
>
> Yes, we find that the peak at K=5 was indeed an artifact of the specific dataset and architecture that we were ablating.  Since submission, we have conducted a very detailed ablation of K across multiple datasets and architectures, we empirically find that not choosing topK, gives the best results on average. Currently all our reported results use K=5, therefore we expect our numbers to increase by a few decimals after incorporating this change. We subsample 1 image per category for the ablation study.
>
> | ConvNext-Tiny | ImageNet | ImageNet-R |
> | :---      |   :----:   |  :----:   |
> | K=2 | 82.6  | 47.5 |
> | K=5 | 82.9  | 45.5 |
> | K=50 | 82.6  | 49 |
> | No K | **83.0**  | **51** |
>
> | ResNet18 | ImageNet | ImageNet-R |
> | :---      |   :----:   |  :----:   |
> | K=2 | 73.5  | 37 |
> | K=5 | 77.3  | 41 |
> | K=50 | **78.2** | **43** |
> | No K | 78.2  | 42.5 |
>
> | CLIP-ViT/L-14 | FGVC | Pets |
> | :---      |   :----:   |  :----:   |
> | K=1 | 21  | 79.3 |
> | K=5 | 21  | 79.3 |
> | K=50 |  21  | 79.3 |
> | No K | 21  | 79.3 |
>
> **Q2.4: “this is the first work that adapts pre-trained large-scale discriminative models to individual images” is wrong, TPT also adapts large scale pre-trained classifiers. Add TTA baselines on the Open-Vocabulary CLIP experiment.**
>
> Thank you for pointing out the TPT method. We were not aware of the method. We will remove the sentence and add TPT to the related work.
> We present comparisons with TPT on zero-shot classification in the following table.  For TPT baseline, we tune the prompt and evaluate classification on each dataset.
>
> | | Food101 | CIFAR100 | FGVC | Pets | Flower102 | ImageNet | Average Improvement |
> | :---      |   :----:   |  :----:   |  :----:   |  :----:   |  :----:   |   :----:   |     :----:   |
> | **CLIP-ViT-B/32** | 78.4 | 60.0 | 18.8 | 77.8 | 64.1 | 56.3 ||
> | $~~~~$+TPT | 79.8 (+1.4) | **65.2 (+5.2)** | 18.4 (-0.4) | 77.3 (-0.5) | 62.8 (-1.4) | 57.5 (+1.2) | +0.91 |
> | $~~~~$+Diffusion-TTA | **80.2 (+1.8)** | 61.8 (+1.8) | **22.2 (+3.4)** | **81.1 (+3.3)** | **64.3 (+0.2)** | **58.1 (+1.8)** | **+2.01** |
> | **CLIP-ViT-B/16** | 84.7 | 68.8 | 21.4 | 80.5 | 67.6 | 60.6 | |
> | $~~~~$+TPT | 85.2 (+0.5) | **68.4 (-0.4)** | 20.4 (-1.0) | 80.0 (-0.5) | **70.0 (+2.4**) | **61.6 (+1.0)** | +0.31 |
> | $~~~~$+Diffusion-TTA | **85.5 (+0.8)** | 67.6 (-0.8) | **22.6 (+1.2)** | **80.8 (+0.3)** | 69.2 (+1.6) | 61.5 (+0.9) | **+0.67** |
> | **CLIP-ViT-L/14** | 91.2 | 79.6 | 29.0 | 89.2 | 75.2 | 68.9 | |
> | $~~~~$+TPT | 90.6 (-0.6) | 80.6 (+1.0) | 30.2 (+1.2) | 87.6 (-1.6) | **76.9 (+1.6)** | **70.6 (+1.7)** | +0.55 |
> | $~~~~$+Diffusion-TTA | **91.2 (0.0)** | 80.6 (+1.0) | **30.6 (+1.6)** | **89.8 (+0.6)** | 76.1 (+0.9) | 69.9 (+1.0) | **+0.85** |
>
>
> We conclude that:
>
> 1. Our Diffusion-TTA consistently outperforms TPT  across all CLIP backbone architectures.
> 2. TPT only adapts prompts and is thus restricted to vision-language classifiers like CLIP.  Our method instead can improve any pre-trained classifier, and even dense pixel labeller (e.g., semantic segmentation)
> 3. Our method works for both single-example and online test-time adaptation, whereas TPT only works on single-example settings.
>
>
> **Q2.5: Writing Suggestions**
>
> Thank you for your suggestions.  We have revised our paper based on your suggestions.

---

> > ### Comment · Reviewer_UVNj · 2023-08-19
> > **Response to authors’ rebuttal**
> >
> > Thanks for providing very detailed responses to my concerns. Since most of my concerns have been addressed, I'm happy to raise my rating to weak accept. I hope the authors can incorporate all the changes and new results into the revised version of the paper.

---

> > > ### Author Response · Authors · 2023-08-19
> > >
> > > Thanks for suggesting the additional TPT baseline, it help us improve our work. As suggested by the reviewer, we will include the changes and new results in the final paper.

---

### Author Rebuttal · Authors · 2023-08-10

Thank you for your constructive feedback and for spending the time to read our paper in detail.

Below we address the common concern of all the reviewers:

**Q1.1: Use same backbones when comparing against TENT and CoTTA baselines.**

We re-evaluate COTTA and TENT using the same backbone as ours.

Note that, TTT-MAE builds atop customized classifiers.  Therefore we cannot evaluate TTT-MAE with the same classifiers without re-training it on the ImageNet dataset.

We evaluate Diff-TTA / TTT-MAE under single-example settings, and CoTTA / TENT under online settings.  COTTA and TENT are not applicable for single-example settings: they explicitly require pseudo-labels (entropy minimization) to improve the classification accuracy.

||ImageNet|ImageNet-A|ImageNet-R|ImageNet-C|ImageNet-V2|
|:---|:----:|:----:|:----:|:----:|:----:|
|**Customized ViT-L/16 backbone**|82.1|14.4|33.0|17.5|72.5|
|$~~~~$+TTT-MAE (single-sample)|82.0 (-0.1)|21.3 (+6.9)|39.2 (+6.2)|27.5 (+10.0)|72.3 (-0.2)|
|**ResNet18**|69.5|1.4|34.6|2.6|57.1|
|$~~~~$+TENT (online)|63.0 (-6.5)|0.6 (-0.8)|34.7 (+0.1)|**12.1 (+9.5)**|52.0 (-5.1)|
|$~~~~$+CoTTA (online)|63.0 (-6.5)|0.7 (-0.7)|34.7 (+0.1)|11.7 (+9.1)|52.1 (-5.0)|
|$~~~~$+Diffusion-TTA (single-example)|**77.2 (+1.7)**|**6.1 (+4.7)**|**39.7 (+5.1)**|4.5 (+1.9)|**63.8 (+6.7)**|
|**ViT-B/32**|75.7|9.0|45.2|39.5|61.0|
|$~~~~$+TENT (online)|75.7 (0.0)|9.0 (0.0)|45.3 (+0.1)|38.9 (-0.6)|61.1 (+0.1)|
|$~~~~$+CoTTA (online)|75.8 (+0.1)|8.6 (-0.4)|45.0 (-0.2)|40.0 (+0.5)|60.9 (-0.1)|
|$~~~~$+Diffusion-TTA (single-example)|**77.6 (+1.9)**|**11.2 (+2.2)**|**46.5 (+1.3)**|**41.4 (+1.9)**|**64.4 (+3.4)**|
|**ConvNext-Tiny**|81.9|22.7|47.8|16.4|70.9|
|$~~~~$+TENT (online)|79.3 (-2.6)|10.6 (-12.1)|42.7 (-5.1)|2.7 (-13.7)|69.0 (-1.9)|
|$~~~~$+CoTTA (online)|80.5 (-1.4)|13.2 (-9.5)|47.2 (-0.6)|13.7 (-2.7)|68.9 (-2.0)|
|$~~~~$+Diffusion-TTA (single-sample)|**83.1 (+1.2)**|**25.8 (+3.1)**|**49.7 (+1.9)**|**21.0 (+4.6)**|**71.5 (+0.6)**|

We conclude:

- Diff-TTA outperforms TENT and CoTTA across various architecture backbones and distribution shifts.

- Diff-TTA outperforms TTT-MAE even with a smaller classifier.  ConvNext (28.5M parameters) optimized by Diffusion-TTA (online) achieves better performance than a much bigger custom backbone of ViT-L + ViT-B (392M parameters) optimized by TTT-MAE (online)

---
Since submission we have extended the proposed method in two ways:

**Q1.2: Online Test-Time Adaptation:**

We have extended our method to an online TTA setting:  We adapt model weights to a set of streaming examples without resetting  the model weights at each input example. We compare our method to TTT-MAE, CoTTA, and TENT for online adaptation.

|ImageNet Corruption:|Gaussian-Noise|$~~~$Fog|Pixelate|Snow|Contrast|
|:---  |  :----:| :----:| :----:| :----:| :----:|
|**Customized ViT-L/16 classifier**|17.1|38.7|47.1|35.6|6.9|
|$~~~~$+TTT-MAE (single-sample)|27.9 (+10.8)|45.1 (+6.4)|61.4 (+14.3)|43.2 (+7.6)|9.3 (+2.4)|
|$~~~~$+TTT-MAE (online)|37.9 (+20.8)|51.1 (+12.4)|65.7 (+18.6)|56.5 (+20.9)|10 (+3.1)|
|**ResNet50**|6.3|25.2|26.5|16.7|3.6|
|$~~~~$+TENT (online)|12.3 (+6.0)|43.2 (+18.0)|41.8 (+15.3)|28.4 (+11.7)|**12 (+8.4)**|
|$~~~~$+CoTTA (online)|12.2 (+5.9)|42.4 (+17.2)|41.7 (+15.2)|28.6 (+11.9)|11.9 (+8.3)|
|$~~~~$+Diffusion-TTA (single-sample)|12.7 (+6.4)|33.0 (+7.8)|30.4 (+3.9)|28.7 (+12.0)|7.0 (+3.4)|
|$~~~~$+Diffusion-TTA (online)|**19 (+12.7)**|**43.2 (+18.0)**|**50.2 (+23.7)**|**33.6 (+16.9)**|2.7 (-0.9)|
|**ViT-B/32**|39.5|35.9|55|30|31.5|
|$~~~~$+TENT (online)|38.9 (-0.6)|35.8 (-0.1)|55.5 (+0.5)|30.7 (+0.7)|32.1 (+0.6)|
|$~~~~$+CoTTA (online)|40.0 (+0.5)|34.6 (-1.3)|54.5 (-0.5)|29.7 (-0.3)|32 (+0.5)|
|$~~~~$+Diffusion-TTA (single-sample)|41.4 (+1.9)|47.3 (+11.4)|43.5 (-11.5)|36.1 (+6.1)|20.8 (-10.7)|
|$~~~~$+Diffusion-TTA (online)|**46.5 (+7.0)**|**56.2 (+10.3)**|**64.7 (+9.7)**|**50.4 (+20.4)**|**33.6 (+2.1)**|
|**ConvNext-Tiny**|16.4|32.3|37.2|38.3|32|
|$~~~~$+TENT (online)| 2.7 (-13.7)|5 (-27.3)|43.9 (+6.7)|15.2 (-23.1)|40.7 (-18.7)|
|$~~~~$+CoTTA (online)|13.7 (-2.7)|29.8 (-2.5)|37.3 (+0.1)|26.6 (-11.7)|32.6 (+0.6)|
|$~~~~$+Diffusion-TTA (single-sample)|21.0 (+4.6)|46.9 (+14.6)|41 (+3.8)|45.6 (+7.3)|25 (-7.0)|
|$~~~~$+Diffusion-TTA (online)|**47.4 (+31.0)**|**65.9 (+33.6)**|**69 (+31.8)**|**62.6 (+24.3)**|**46.2 (+14.2)**|

From the table we see that:
- Online adaptation of Diff-TTA significantly outperforms the single-example setting of Diff-TTA
- Diff-TTA outperforms TENT and CoTTA across various backbones. CoTTA and TENT fail to improve classification on certain corruptions and classifiers.  Our results are consistent with the analysis in [1]: the authors find that methods that primarily work under online settings (TENT or CoTTA) are not robust to different types of architectures and distribution shifts (as reported in Table 4 of [1]).
- Our method outperforms TTT-MAE even with a smaller-size classifier.  ConvNext (28.5M params) optimized by Diffusion-TTA (online) achieves better performance than a much bigger custom backbone of ViT-L + ViT-B (392M params) optimized by TTT-MAE (online)

[1] On the Pitfalls of test-time adaptation. Zhao et al

**Q1.3: Diff-TTA for Semantic Segmentation:**

We have extended our method to semantic segmentation tasks by replacing the image classifier with a pre-trained segmentor.  See Figure 1 and 4 in the attached pdf for the architecture diagram and qualitative results.

Both the segmentor and the diffusion model are trained on ADE20K dataset. Under single-example settings, we find Diff-TTA consistently improves pre-trained SegFormer.


| ADE20K Corruption: | Clean | Gaussian-Noise | $~~$Fog | Frost | Snow | Contrast | Shot
| :---      | :----:   | :------:        | :----:   | :----:   |   :----:   |  :----: |  :----: |
| **SegFormer** | 66.1 | 65.3 | 63.0 | 58.0 | 55.2 | 65.3 | 63.3
| $~~~~$+Diffusion-TTA | 66.1 | **66.4 (+1.1)** | **65.1 (+2.1)** | **58.9 (+0.9)** | **56.6 (+1.4)** | **66.4 (+1.1)** | **63.7 (+0.4)**

---

### Decision · Program_Chairs · 2023-09-21

**Decision:**

Accept (poster)

**Comment:**

This work harnesses generative diffusion models as a loss for test-time adaptation to update image classifiers and improve robustness to shift. The subject of this submission, test-time adaptation, is now a popular topic for robustness, and this submission contributes its own approach that is relevant to the current era of large-scale generative modeling: the proposed Diff-TTA method. All five expert reviewers with backgrounds on diffusion and adaptation agree on acceptance (4gPG : 7, qddk: 7, UVNj: 6, 6xj6: 6, VyeN: 6). The authors provide a rebuttal, and then reviewers maintained or raised their scores (UVNj, 6xj6, qddk), because of the experimental results provided. More specifically, the rebuttal included more comprehensive and controlled experiments (with matched backbones, more ablations, and more settings + tasks like open vocabulary and semantic segmentation). The experiments cover a breadth of benchmarks and baselines to show reliable improvement (at the cost of more computation time). The AC sides with acceptance: all reviewers agree, the rebuttal only improved the submission, and this work connects the active area of test-time adaptation with the rapidly-advancing topic of diffusion modeling.

The negatives emphasized by reviewers are novelty (6xj6: Li et al. [22], UVNj: TPT) and computation time (4gPG, 6xj6, qddk). For novelty, the AC acknowledges Li et al. [22], but considers the update in this work to be distinct, and finds it justified by experiments (including the ablation of the choice of parameters in the rebuttal) For test-time adaptation to single images, TPT and the original TTT by Sun et al. ICML'20 are indeed relevant. For computation time, the time needed is clearly mentioned in the implementation details, and the increased time follows from the use of large discriminative + generative models that are the focus of this work. The computation needed distances this work from the common emphasis for test-time adaptation on efficient updates, and may limit applicability, but this work nevertheless provides an informative "proof of principle" (4gPG, in post-rebuttal discussion). As a further plus, Diff-TTA improves in the online setting over the episodic setting (as shown in the rebuttal), as many TTA methods do (and should, one can argue), so perhaps future work could amortize some of the computation across samples.

The AC advises the authors to incorporate the reviewers' feedback, and in particular to include the experiments on the suggested backbones (both for the controlled comparisons and on ResNet-50 as a common model for benchmarking). To more comprehensively connect to research on test-time adaptation, the AC would also like to point out the use of diffusion models for adaptation of inputs (Diffusion for Adversarial Purification. Nie et al. ICML'22; Diffusion-Driven Adaptation. Gao et al. CVPR'23), although it is of course up to the authors to decide if these are sufficiently related to their work.